# Oct4 mediates Müller glia reprogramming and cell cycle exit during retina regeneration in zebrafish

Poonam Sharma, Shivangi Gupta*, Mansi Chaudhary*, Soumitra Mitra*, Bindia Chawla, Mohammad Anwar Khursheed, Rajesh Ramachandran

Octamer-binding transcription factor 4 (Oct4, also known as Pou5F3) is an essential pluripotency-inducing factor, governing a plethora of biological functions during cellular reprogramming. Retina regeneration in zebrafish involves reprogramming of Müller glia (MG) into a proliferating population of progenitors (MGPCs) with stem cell–like characteristics, along with up-regulation of pluripotency-inducing factors. However, the significance of Oct4 during retina regeneration remains elusive. In this study, we show an early panretinal induction of Oct4, which is essential for MG reprogramming through the regulation of several regeneration-associated factors such as Ascl1a, Lin28a, Sox2, Zeb, E-cadherin, and various miRNAs, namely, *let-7a, miR-200a/miR-200b,* and *miR-143/miR-145.* We also show the crucial roles played by Oct4 during cell cycle exit of MGPCs in collaboration with members of nucleosome remodeling and deacetylase complex such as Hdac1. Notably, Oct4 regulates Tgf-β signaling negatively during MG reprogramming, and positively to cause cycle exit of MGPCs. Our study reveals unique mechanistic involvement of Oct4, during MG reprogramming and cell cycle exit in zebrafish, which may also account for the inefficient retina regeneration in mammals.

## Introduction

Tissue regeneration is a complex phenomenon in which the damaged part of the organ is restored to normalcy through a series of genetic and epigenetic transformations of cells near the injury site. Regenerative capability is often limited in the nervous tissue of mammals, compared with their hair, skin, or liver. Interestingly, vertebrates such as fishes and frogs possess remarkable regenerative potential in almost all organs (Gemberling et al, 2013). One of the well-characterized model organisms, zebrafish, has been extensively used to unravel molecular mechanisms underlying tissue

regeneration in general and retina in particular (Goldman, 2014). Upon injury, the Müller glia (MG) cells of the retina reprogram themselves to give rise to MG-derived progenitor cells (MGPCs), which are capable of differentiating into various retinal cell types and MG, as confirmed by lineage tracing (Bernardos et al, 2007; Ramachandran et al, 2010b, 2012a). In comparison with zebrafish, mammalian MG often fails to elicit an adequate regenerative response to restore vision. It is believed that the retina being part of the central nervous system has an inhibitory environment regarding the growth of new nervous tissue. This scenario makes it really interesting to explore how the zebrafish central nervous system is capable of regeneration after an acute injury. Several studies, characterizing various molecular events with special reference to transcription factors, cell signaling networks, epigenome modification, etc., have revealed the complex nature of zebrafish retina regeneration (Goldman, 2014; Gorsuch & Hyde, 2014; Wan & Goldman, 2016). Many of such regeneration-associated gene expression events were missing or inadequate in the injured mammalian retina, which may account for lack of complete retina regeneration in them (Wilken & Reh, 2016). Moreover, artificial induction of some of these transcription factors such as Ascl1a has paved way for improved regenerative response in the injured retina of mice (Brzezinski et al, 2011; Jorstad et al, 2017). However, lack of adequate regenerative response in mammalian models necessitates a deeper investigation into the MG reprogramming of zebrafish retina, which would enable us to connect the missing links of the ever-enigmatic regeneration cascade.

Cellular reprogramming, leading to the induction of progenitors that are capable of regeneration because of their stem cell–like properties, is a wonderful alternative to fibroblast-mediated wound closure and scar formation. Zebrafish retina adopts a plethora of mechanisms that trigger an effective regenerative response (Goldman, 2014; Gorsuch & Hyde, 2014; Wan & Goldman, 2016). The advent of knowledge about the induction of pluripotency in fibroblasts, mediated through pluripotency-inducing factors (PIFs), prompted us to look closely into their molecular functions in context to zebrafish retina regeneration where almost all PIFs are expressed soon after an acute injury (Ramachandran et al, 2010a;

Indian Institute of Science Education and Research, Mohali, India

Correspondence: rajeshra@iisermohali.ac.in
*Shivangi Gupta, Mansi Chaudhary, and Soumitra Mitra contributed equally to this work.

Gorsuch et al, 2017). Although many of them such as Lin28a (Ramachandran et al, 2010a), Sox2 (Gorsuch et al, 2017), and Mycb (Mitra et al, 2019) are characterized previously, the importance of Oct4 still remains undetermined. Oct4 is a homeodomain-containing transcription factor essential for the formation and maintenance of pluripotent stem cells (Nichols et al, 1998). Oct4 is also known to carry out diverse biological functions in embryonic stem cells, cancer cells, and epithelial to mesenchymal transition (EMT) (Radzisheuskaya & Silva, 2014). Oct4 also mediates transcriptional repression through nucleosome remodeling and deacetylase (NuRD) complex during differentiation (Hu & Wade, 2012). Importantly, Oct4 down-regulates the components of Tgf-$\beta$ signaling to facilitate cellular reprogramming in different physiological conditions (Li et al, 2010; Radzisheuskaya & Silva, 2014). It is also important to note that (i) efficient induction of pluripotency necessitates a very high level of Oct4 (Nagamatsu et al, 2012; Polo et al, 2012) and (ii) its expression levels can switch the fate of embryonic stem cells (Radzisheuskaya et al, 2013).

In this study, we explored the significance of the panretinal induction of Oct4 soon after the injury and its interrelationship with Tgf-$\beta$ signaling and other gene expression events at different phases of retina regeneration. We found unique dual roles of Oct4 during MG reprogramming in zebrafish. Furthermore, we demonstrate the significance of the contrasting role of Oct4-mediated signaling events towards the later stages, which is necessary for the cell cycle exit of MGPCs that paves way for complete regenerative response in the zebrafish retina.

## Results

### Oct4 is rapidly induced during zebrafish retina regeneration

The significance of Oct4 is well known to induce pluripotency in human and mouse fibroblasts (Lowry et al, 2008; Li et al, 2010; Radzisheuskaya & Silva, 2014; Chen et al, 2016). Oct4 is also considered a single factor capable of executing a multitude of functions during cellular reprogramming and mesenchymal to epithelial transition (MET) (Radzisheuskaya & Silva, 2014). Its induction during zebrafish retina regeneration is documented with limited information about its regulation (Ramachandran et al, 2010a). Here, we injured zebrafish retina by focal stab using a 30G needle. The oct4 mRNA levels were analyzed after retinal injury by RT-PCR and qRT-PCR (Fig 1A), which showed a double peak in its expression pattern. The first one is at 16 hours posti-njury (hpi), and the second at 4 days post-injury (dpi). The Oct4 levels also showed a similar trend in Western blot analysis of its protein isolated from total retinal extracts at various times post-injury (Fig 1B). Further analysis by mRNA in situ hybridization (ISH) revealed that oct4 mRNA is expressed negligibly in the uninjured retina followed by a panretinal induction at 16 hpi. Later on, the oct4 expression stayed restricted to the site of injury from 2 to 7 dpi (Fig 1C).

A closer evaluation of the oct4-expressing cells at 4 dpi, a time when the progenitor cell proliferation is at its peak, revealed that the oct4$^+$ cells stay just adjacent to the actively proliferating progenitor cells seen through a BrdU pulse labeling assay (Fig 1D).

Quantitative analysis of the BrdU$^+$ and oct4$^+$ cells revealed that ~10% of the total BrdU$^+$ cells showed oct4 expression and about 12% of oct4$^+$ cells exhibited the presence of BrdU from the pulse labeling (Fig 1E). Similar results were obtained for Oct4 protein expression in 4 dpi retina of 1016tuba1a:GFP transgenic retina wherein the MGPCs showed GFP expression (Fig 1F) (Fausett & Goldman, 2006). These observations suggested the following possibilities: (i) oct4-expressing cells do not proliferate but can direct the neighboring cells to proliferate and (ii) oct4 expression is a post-proliferative phenomenon. To determine which is the real scenario at 4 dpi, retinal sections were used to perform oct4 mRNA ISH, followed by staining with proliferating cell nuclear antigen (PCNA) and BrdU. PCNA has a longer half-life and stays detectable beyond the cell cycle exit (Mandyam et al, 2007; Kimmel & Meyer, 2010; Bologna-Molina et al, 2013). Hence, PCNA could be used as a marker of post-proliferative status as well. Interestingly, we found that almost all PCNA$^+$ cells had the oct4 expression, suggesting the existence of the second possibility (Fig 1G and H). These observations were further confirmed by BrdU pulse labeling along with oct4 mRNA ISH at 2, 4, and 8 dpi (Fig S1A). The quantification revealed that the propensity of BrdU and oct4 co-labeling increased only towards the end of the proliferative phase at 8 dpi when most of the BrdU$^+$ cells were exiting the cell cycle (Fig S1B).

We then decided to explore the expression pattern of oct4 through a cell sorting approach, for which we used 1016tuba1a:GFP transgenic retina. GFP-positive and GFP-negative cells were used to assess the levels of oct4 mRNA both qualitatively and quantitatively (Fig 1I and J). High levels of oct4 mRNA were seen in GFP$^+$ cells, which are similar to the PCNA$^+$ ones (Fig S1C), along with negligible expression in the GFP$^-$ ones. These observations suggested that despite being secluded from actively proliferating MGPCs, the oct4 expression is an immediate and transient feature of post-proliferative cells at 4 dpi.

We further explored the regulatory factors that could influence the expression of oct4 soon after the injury. One of the potential candidates was ascl1a, an essential regeneration-associated gene (Ramachandran et al, 2010a, 2011), which also shows a panretinal early expression soon after injury in zebrafish retina (Ramachandran et al, 2011), similar to oct4. To address whether Ascl1a influences oct4 expression, we checked the levels of oct4 in the ascl1a knockdown background. We found a significant decline in oct4 mRNA levels because of ascl1a knockdown (Fig S1D and E). Analysis of the oct4 promoter revealed several Ascl1a-binding sites (BSs), which were confirmed to be functional in a chromatin immunoprecipation (ChIP) assay done using 16 hpi retinal extract (Fig S1F). These observations suggest that Ascl1a is one of the governing factors that cause the upregulation of Oct4.

### Oct4-mediated gene regulatory network is essential for retina regeneration

We were intrigued by the fact that there is an abundant expression of oct4 mRNA panretinally at 16 hpi, which later on stayed restricted to post-proliferative MGPCs. To assess its significance, we adopted a knockdown approach to eliminate Oct4 soon after the injury using two different lissamine-tagged morpholino (MO)-based antisense oligos targeting oct4 mRNA. We performed the oct4 knockdown

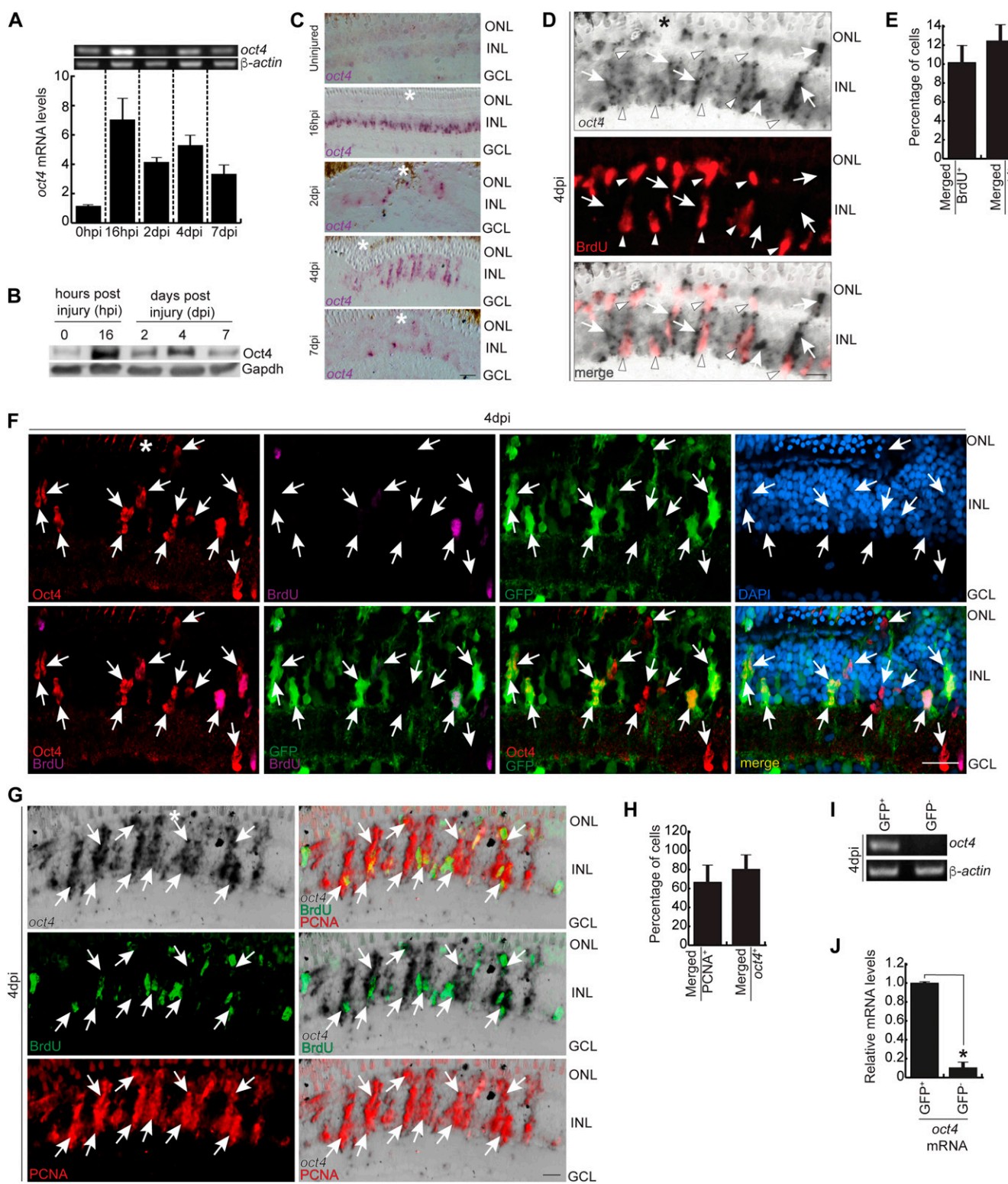

**Figure 1. The expression pattern of Oct4, its association with MGPCs, and seclusion from BrdU⁺ cells.**
**(A)** RT PCR of *oct4* mRNA (upper) and its qRT-PCR (lower) at various time points post retinal injury. **(B)** Western blot analysis of Oct4 from retinal extracts collected at different time points post injury. Gapdh is the loading control. **(C)** Bright-field (BF) microscopy images of retinal cross sections showing the mRNA ISH of *oct4* at various time points post retinal injury. **(D, E)** BF and immunofluorescence (IF) confocal microscopy images of retinal cross section showing the mRNA ISH reveals the *oct4* expression in the neighboring cells of BrdU⁺ MGPCs at 4 dpi (D), which is quantified (E). **(D)** White arrowheads mark BrdU⁺ and *oct4⁻* cells and white arrows mark *oct4⁺* but BrdU⁻ cells in (D). **(F)** IF confocal microscopy images of retinal cross section, which shows the Oct4 immunofluorescence in GFP⁺ MGPCs in 4 dpi retina of *1016tuba1a*:GFP

experiments as per the experimental timeline (Fig 2A). As done with previous experiments, 3 h before harvesting, a BrdU/EdU pulse was given at 4 dpi to assess the number of MGPCs in the *oct4* knockdown scenario. We found a significant decline in the number of BrdU[+]/EdU[+] MGPCs in an *oct4* MO concentration-dependent manner in the retina at 4 dpi (Figs 2B and C, and S2A and B). The negative effect of *oct4* knockdown on MGPC proliferation was rescued by the transfection of *gfp-oct4* fusion mRNA into the retina at the time of injury (Fig S2C and D). Furthermore, the *oct4* knockdown (Fig 2D) caused a decline in both *oct4* mRNA and protein levels, which were confirmed by qRT-PCR (Fig 2E) and Western blot (Fig 2F) analysis of retinal lysates at both 16 hpi and 2 dpi. The *oct4* knockdown also caused a significant decline in regeneration-associated transcription factors Ascl1a (Ramachandran et al, 2010a, 2011) and Sox2 (Gorsuch et al, 2017) in 16 hpi and 2 dpi retina (Fig 2E and F). These observations supported the view that early induction of *oct4* is necessary for the normal regenerative response of the retina.

We further explored the *cis*-regulatory regions of regeneration-associated genes *ascl1a* and *oct4* itself for Oct4-binding consensus sequence ATGCAAAT (Kemler et al, 1991) (Oct4-BS). We found one Oct4-BS on the *ascl1a* promoter and five of them on *oct4* promoter sequences. ChIP assay performed in 16 hpi retinal extract confirmed that Oct4 indeed bound to all the sites in *oct4* and *ascl1a* gene promoter (Fig S3A and B). The mRNA ISHs of *ascl1a* and *oct4* in *oct4* knockdown background in 4 dpi retina also supported the results above (Figs S3C, 2E, and F). The Oct4-mediated transactivation of *ascl1a* promoter is further confirmed by luciferase assay performed in zebrafish embryos co-injected with *ascl1a* promoter driving GFP–luciferase fusion construct along with *oct4*-targeting MO or *oct4* mRNA (Fig S3D and E). These observations suggested that Oct4 and Ascl1a indulge in a mutual positive feedback loop during retina regeneration.

Interestingly, despite the reduction in the number of MGPCs upon *oct4* knockdown, we saw an up-regulated expression of *lin28a* (Fig 2G–I), one of the PIFs essential for normal retina regeneration (Ramachandran et al, 2010a; Kaur et al, 2018). We sought to explore the reasons behind this intriguing finding. The analysis of the *lin28a* promoter did not show any Oct4-BS, which ruled out the possibility of Oct4 directly regulating *lin28a*. We then explored if the *lin28a* could be regulated through Her4.1, an effector of Delta-Notch signaling and transcriptional repressor of *lin28a* in regenerating retina (Mitra et al, 2018). We saw a significant decline in *her4.1* levels in *oct4* knockdown retina (Fig 2G–I), which explained the up-regulated *lin28a* levels. Furthermore, analysis of *her4.1* promoter revealed the presence of Oct4-BS, which was occupied by Oct4 as revealed from ChIP assay performed in 16 hpi retinal extract (Fig 2J). This finding affirms the indispensability of Oct4 in MGPC induction during retina regeneration.

Similarly, we explored whether the decline in BrdU[+] cells found in the retina because of the *oct4* knockdown affected the genes

responsible for cell cycle progression. For this, we analyzed the expression pattern of several proliferation-associated genes such as *cyclins* and *delta* genes in *oct4* knockdown background. The qRT-PCR analysis revealed that cyclin family members *ccna1*, *ccnb1*, *ccnd1*, and *ccne1*, and delta family *dla*, *dlb*, *dlc*, and *dld* indeed were down-regulated in agreement with reduced MGPC proliferation in the absence of Oct4 (Fig S3F). Taken together, these results revealed the potential roles played by Oct4 during MG reprogramming through regulation of *ascl1a*, *sox2*, *her4.1*, and *lin28a* to induce MGPCs.

## Oct4 regulates Tgf-β signaling during retina regeneration

Cellular reprogramming to induce pluripotent stem cells resembles MET in various aspects (Esteban et al, 2012; Shu & Pei, 2014), which involves regulation of genes such as *cdh1* (E-cadherin). One of the important functions governed by Oct4, while acting as a pluripotency inducer, is to activate E-cadherin (Shen et al, 2014) and down-regulate TGF-β signaling (Li et al, 2010). The Tgf-β signaling is known to up-regulate the *snail* family of genes, which repress cellular reprogramming (Li et al, 2010). Snail also functions as a transcriptional repressor of E-cadherin in tumor cells (Batlle et al, 2000). We probed further if *snail* gene family members get up-regulated because of Oct4 knockdown in regenerating retina. We found that four members of *snail* gene family, namely, *snai1a*, *snai1b*, *snai2*, and *snai3* get significantly up-regulated in *oct4* knockdown retina at 2 dpi (Fig 3A and B). Supporting this observation, in contrast to the expression pattern of *oct4* in *1016tuba1a*:GFP transgenic retina (Fig 1J), we found down-regulation of *snail* gene family members in GFP[+] cells compared with rest of the retina (Fig S3G). Furthermore, the *oct4* knockdown up-regulated the Tgf-β signaling components such as *tgfbr1b*, *tgfb2*, and its effector genes *tgfbi* and *smad7* in 2 dpi retina (Fig 3C). Subsequently, we explored if the *oct4* knockdown influenced the expression of *cdh1*, which is important in imparting stemness properties to cells. We saw an increase in the levels of *cdh1* in response to *oct4* knockdown in 2 dpi retina (Fig 3D). In *1016tuba1a*:GFP transgenic retina, we also saw a down-regulation of *cdh1* in GFP[+] cells as compared with the rest of the retina (Fig S3H). In contrast to the previous reports (Redmer et al, 2011), wherein Oct4 activates *cdh1*, which can even replace the requirement of Oct4, we observed the opposite regulation in regenerating retina. Moreover, there was no Oct4-BS on the *cdh1* promoter. Closer analysis of the *cdh1* promoter sequences revealed the presence of BSs of Oct4-regulated transcription factors, namely, Ascl1a and Sox2. Interestingly, Ascl1a and Sox2 bound to their respective BSs, CACCTG (Ramachandran et al, 2010a) and CATTGTA (Mistri et al, 2015) on *cdh1* promoter as revealed in a ChIP assay performed in 16 hpi and 2 dpi retinal extracts (Fig 3E). However, the observed increase in *cdh1* expression because of *oct4* knockdown remained unclear. We speculate that some unknown repressors regulated by Oct4 could potentially mediate the regulation of *cdh1*. These

---

transgenic fish. White arrows mark Oct4[+] and GFP[+] cells. DAPI was used as the counterstain to mark nucleus. **(G, H)** BF and IF confocal microscopy images of retinal cross section show the mRNA ISH of the *oct4* in a significant proportion of PCNA[+] MGPCs at 4 dpi (G), which is quantified (H). **(G)** White arrows mark PCNA[+] cells that are *oct4*[+] in (G). **(I, J)** RT-PCR (I) and qRT-PCR (J) of *oct4* mRNA from GFP[+] MGPCs compared with the GFP[−] cells present in rest of the retina from *1016tuba1a*:GFP transgenic fish at 4 dpi, *P < 0.003 (*t* test), N = 12. Error bars are SD. **(C, D, F, G)** Scale bars, 10 μm; the asterisk marks the injury site; GCL, ganglion cell layer; INL, inner nuclear layer; ONL, outer nuclear layer (C, D, F, G).

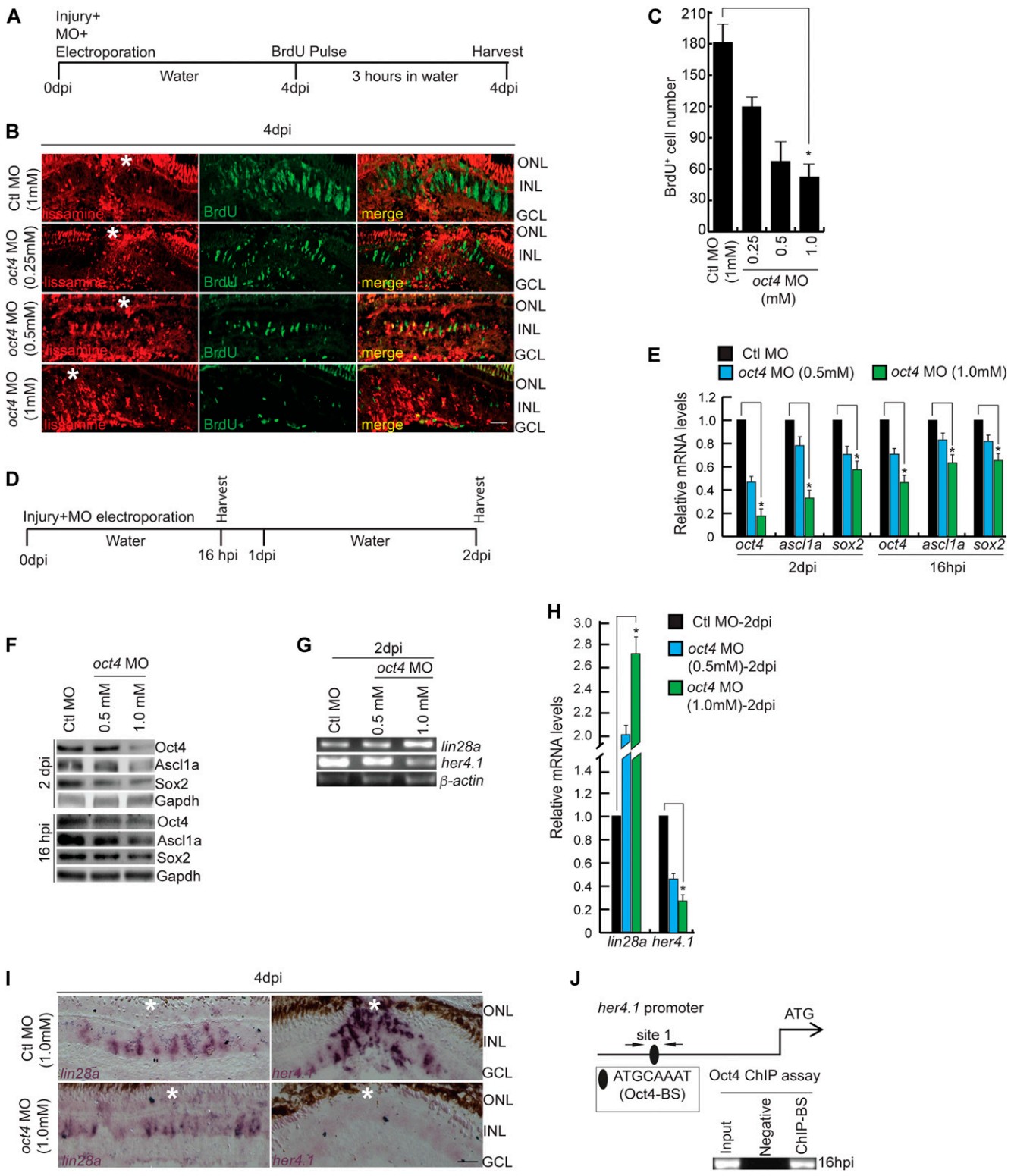

**Figure 2. Oct4 is essential during retina regeneration.**
**(A)** An experimental timeline that describes the MO delivery, electroporation, and BrdU pulse before harvesting at 4 dpi. **(B, C)** IF confocal microscopy images of retinal cross sections show the decline in BrdU⁺ MGPCs with increasing concentrations of *oct4* MO (lissamine tag) at 4 dpi (B), which is quantified (C); *P < 0.0001 (*t* test), N = 4. **(D)** An experimental timeline that describes the MO delivery, electroporation, and harvest at 16 hpi and 2 dpi. **(E)** The qRT-PCR analysis of *oct4*, *ascl1a*, and *sox2* genes in *oct4* knockdown retina at 2 dpi and 16 hpi; *P < 0.01 (*t* test), N = 4. **(F)** Western blot analysis of Oct4, Ascl1a, and Sox2 from retinal extracts collected after *oct4* knockdown at 16 hpi and 2 dpi. Gapdh is the loading control. **(G, H)** RT-PCR (G) and qRT-PCR (H) of *lin28a* and *her4.1* in *oct4* knockdown retina at 2 dpi. **(I)** BF microscopy images of

results are suggestive of the existence of a functional Oct4/Ascl1a/Sox2/E-cadherin and Oct4/Tgf-β signaling/Snail regulatory axes necessary for the formation of MGPCs during retina regeneration.

## Oct4 influences *miR-200*/Zeb regulatory loop during regeneration

Tissue reprogramming during regenerative response often involves a fine balance among various transcription factors and oscillations between EMT and MET (Liu et al, 2013; Forte et al, 2017). The initial phase of induced pluripotency in fibroblasts is similar to MET (Li et al, 2010). The zinc finger E-box–binding homeodomain transcription-repressing factors, ZEB1 and ZEB2, are necessary for normal development in vertebrates (Gheldof et al, 2012). ZEB1 also mediates EMT, the opposite phenomenon of MET, through transcriptional repression of E-cadherin (Chua et al, 2007; Peinado et al, 2007; Sanchez-Tillo et al, 2010; Schulte et al, 2012; Galvan et al, 2015; Zhang et al, 2015) and members of *miR-200* family (Wellner et al, 2009). To decipher whether similar pathways exist during retina regeneration, we explored the changes in the levels of *zeb* genes in response to retinal injury and *oct4* knockdown. The *zeb1a* and *zeb2a* expression pattern in postinjured retina revealed an immediate early induction (Fig S3I), suggestive of their significance in early MG reprogramming during retina regeneration. With *oct4* knockdown, the levels of *zeb1a*, *zeb1b*, *zeb2a*, and *zeb2b* were significantly down-regulated in a concentration-dependent manner in 2 dpi retina (Fig 3A and F). Further analysis of the *zeb* gene promoters revealed the existence of typical Oct4-BSs, which were occupied by the endogenous Oct4 as confirmed by ChIP assay performed using 16 hpi retinal extract (Fig 3G). Interestingly, analysis of mRNA levels of *zeb* gene family members revealed an elevated expression levels in GFP⁻ cells than the GFP⁺ ones sorted from *1016tuba1a*:GFP transgenic retina (Fig S3J). Based on these observations, we speculated if the Zeb transcriptional repressors could be responsible for Oct4-mediated *cdh1* repression (Fig 3D) and inhibition of *oct4* itself. To examine this, we transfected varying concentrations of *zeb1a* and *zeb2a* mRNA in postinjured retina and explored the levels of *cdh1* and *oct4*. We found *zeb1a* (Fig 3H) and *zeb2a* (Fig S3K) mRNA dose-dependent decline in the levels of both *cdh1* and *oct4* in 16 hpi and 2 dpi retina. These results suggest that Oct4-mediated regulation of *cdh1* could be mediated through Zeb1a/Zeb2a, and considering the restricted expression pattern of *oct4* in the 4 dpi retina, we presume that Zeb1a/Zeb1b transcriptional repressors play a role in restricting the early panretinal expression of *oct4* to the site of injury through a mutual regulatory relationship.

Next, we explored if the *miR-200*/Zeb1 axis contributed to the Oct4-mediated MG reprogramming. Zeb1 plays a transcriptional repressive role on *miR-200* promoter during zebrafish development (Vannier et al, 2013). To ascertain this, we checked the levels of *miR-200a* and *miR-200b*, the translational repressors of *zeb* mRNAs (Park et al, 2008), in response to injury as well as *oct4* knockdown in 2 dpi retina. We found an up-regulation of *miR-200a* and *miR-200b* soon after injury (Fig S3L), and surprisingly further high levels of *miR-200a* and *miR-200b* in 2 dpi retina after *oct4* knockdown (Fig 3I).

Similar to *miR-200* family, both *miR-143* and *miR-145* had an immediate early up-regulation soon after retinal injury (Fig S3M) and high levels in *oct4* knockdown retina (Fig 3I). The *miR-143*/*miR-145* are inhibitors of stem cell characteristics and are also the translational repressors of PIFs such as *oct4*, *cmyc*, and *klf4* mRNAs (Huang et al, 2012). In support of these observations, expression analysis of these miRNA genes revealed higher dose of expression in GFP-negative than observed in GFP-positive cells sorted from *1016tuba1a*:GFP transgenic retina (Fig S3N). These results suggested the existence of a potential repressive mechanism on these miRNA promoters mediated through Oct4 in the retina. ChIP assay performed in 16 hpi retinal extract using anti-Oct4 antibody confirmed the binding of Oct4 on the promoters of *miR-200a*/*miR-200b* (Fig 3J) and *miR-143*/*miR-145* (Fig 3K). We speculated that the repressive role of Oct4 binding on *miR-200a*/*miR-200b* and *miR-143*/*miR-145* may be carried out in collaboration with repressive factors such as histone deacetylase1 (Hdac1), as reported in embryonic stem cells (van den Berg et al, 2010). The retinal ChIP assay performed using anti-Hdac1 antibody confirmed that it could bind to the Oct4-BS of both *miR-200a*/*miR-200b* and *miR-143*/*miR-145* promoters (Fig 3J and K). To validate these observations, we performed a co-immunoprecipitation (Co-IP) assay in 16 hpi retinal extracts using both anti-Oct4 and anti-Hdac1 antibodies in separate experiments and probed for both these proteins. Interestingly, we found Oct4-Hdac1 interactions as revealed by the Co-IP assay (Fig 3L and M). These results suggest that the Oct4 causes transcriptional activation of *zeb* mRNAs and repression of *miR-200a*/*miR-200b* and *miR-143*/*miR-145*.

## Snails, Zebs, *miR200a*/*miR-200b*/*miR-143*/*miR-145*, and Cdh1 regulates the number of MGPCs during regeneration

Our study demonstrated the importance of Oct4 in regulating various factors such as Snails, Zebs, *miR200a*/*miR-200b*/*miR-143*/*miR-145*, and E-cadherin which are known to contribute to cellular reprogramming. However, the importance of these factors during retina regeneration remained obscure. To explore this, we decided to adopt a gene overexpression/knockdown approach in a context-dependent manner and also along with *oct4* MO when required. As we demonstrated a decline in MGPC proliferation with *oct4* knockdown (Fig 2B and C), along with an up-regulation of *snail* gene family members (Fig 3A and B), we speculated that Snails might have important roles to play during retina regeneration. To address this question, we transfected the injured retina using *snai1a*, *snai1b*, *snai2*, and *snai3* mRNAs in separate experiments. Interestingly, we found a significant decline in the number of MGPCs in retina transfected with *snail* gene family members in a concentration-dependent manner (Fig 4A and B). These results suggest the anti-proliferative effect of Snails during MGPCs proliferation.

Similarly, we found a decline in the expression levels of *zeb* gene family members (Fig 3A and F) because of *oct4* knockdown. To ascertain if the *oct4* knockdown-mediated decline in MGPCs could be

---

retinal cross sections show the expression of *lin28a* and *her4.1* mRNA in *oct4* knockdown retina at 4 dpi. **(J)** The *her4.1* promoter schematic reveals the typical Oct4-BS (upper) and the retinal ChIP assay confirms the physical binding of Oct4 at the typical BS (lower) in 16 hpi retina. Ctl MO is control MO. Error bars are SD. **(B, I)** Scale bars, 10 μm; the asterisk marks the injury site; GCL, ganglion cell layer; INL, inner nuclear layer; ONL, outer nuclear layer (B, I).

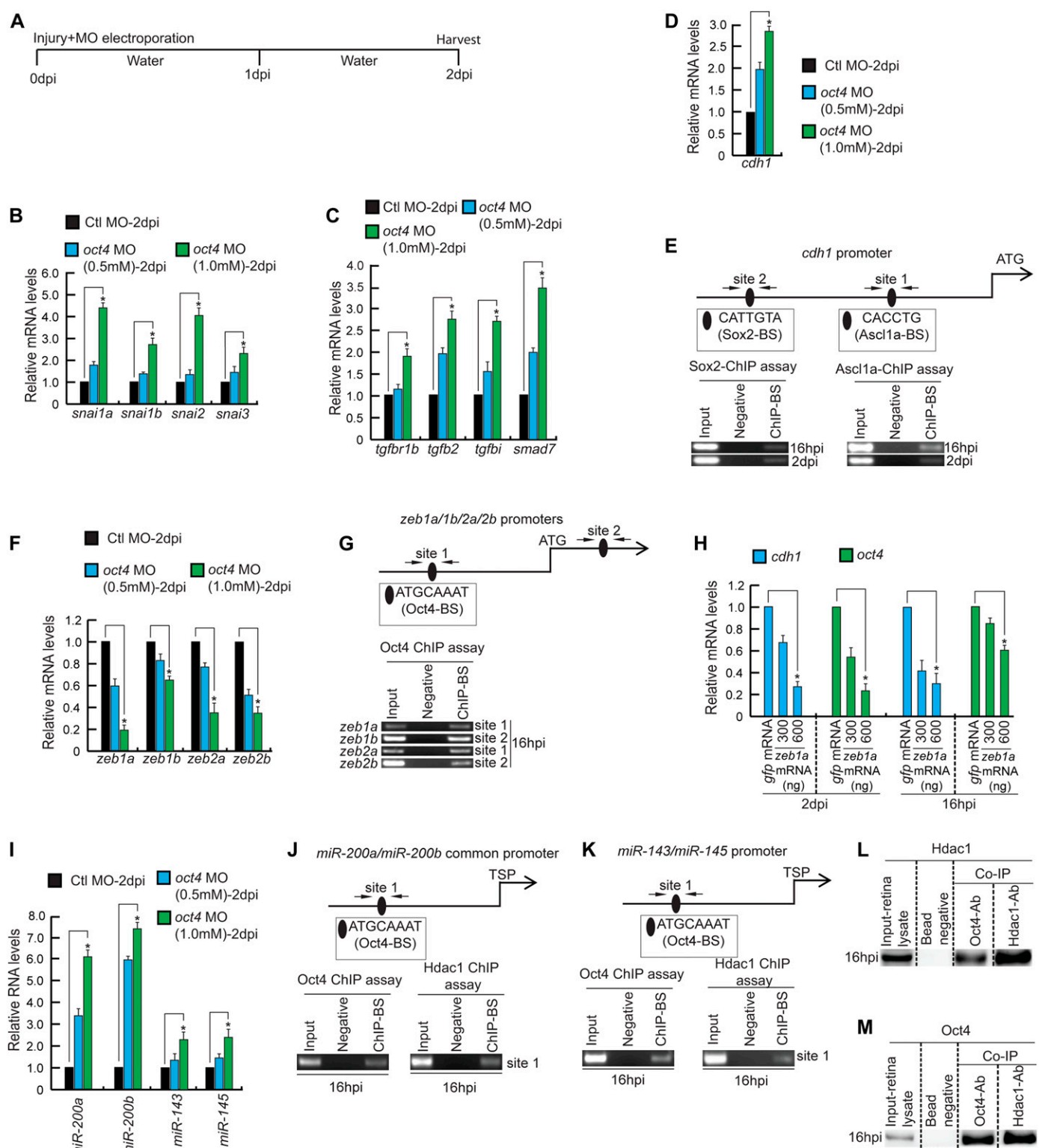

**Figure 3. The importance of Oct4 during retina regeneration revealed through various gene regulatory events.**
**(A)** An experimental timeline that describes the MO delivery, electroporation, and retina harvest at 2 dpi. **(B)** The qRT-PCR analysis of *snail* family genes in *oct4* knockdown retina at 2 dpi; *P < 0.004 (*t* test), N = 4. **(C)** qRT-PCR analysis of various component genes of Tgf-β signaling in *oct4* knockdown retina at 2 dpi; *P < 0.003 (*t* test), N = 4. **(D)** qRT-PCR analysis of *cdh1* mRNA in *oct4* MO-electroporated retina at 2 dpi; *P < 0.01 (*t* test), N = 4. **(E)** The *cdh1* promoter schematic reveals the Sox2 and Ascl1a BSs (upper), and the retinal ChIP assays confirm the physical binding of Sox2 and Ascl1a (lower) in 16 hpi and 2 dpi retina. **(F)** qRT-PCR analysis of *zeb* family genes in *oct4* knockdown retina at 2 dpi; *P < 0.02 (*t* test), N = 4. **(G)** The promoter and first intron schematic (upper) of *zeb* family genes reveal the presence of

alleviated by Zeb overexpression, we adopted an experimental strategy in which *gfp* mRNA and *zeb1a*/*zeb2a* mRNA were transfected along with control or *oct4* MO in separate experiments. These experiments were aimed at finding the influence of replenishment of Zeb in *oct4* knockdown retina and also the effect of Zeb overexpression in control MO-electroporated conditions. Interestingly, at 4 dpi, we found that overexpression of *zeb1a* (Fig 4C and D) and *zeb2a* (Fig S4A and B) had an anti-proliferative effect on MGPCs irrespective of whether control or *oct4* MO were electroporated in the injured retina. It is also important to note in this context that Oct4 is expressed in GFP⁺ MGPCs (Fig 1J) and the *zeb* gene family is expressed at higher levels in non-proliferating GFP⁻ cells (Fig S3J) of *1016tuba1a:gfp* transgenic retina. These results suggest that Oct4 activates the *zeb* gene family to keep the proliferation within the desired limits.

Furthermore, we explored the significance of Oct4-mediated regulation of *miR-200a*, *miR-200b*, *miR-143*, and *miR-145* genes during retina regeneration. We already demonstrated that Oct4 has a suppressive role on the expression of these miRNAs (Fig 3I–K) and an associated decline in MGPC proliferation. Furthermore, these four miRNAs showed a drastic up-regulation soon after a retinal injury (Fig S3L and M). In this scenario, we decided to knockdown these four miRNAs individually and probe for its influence on MGPC proliferation. Interestingly, knockdown of *miR-200a* and *miR-200b* caused a profound increase in MGPCs (Fig 4E and F), whereas the opposite was seen with *miR-143* and *miR-145* knockdown (Fig 4G and H). These observations suggest the significant roles played by Oct4 in causing a definitive number of MGPCs at the site of injury, through the regulation of these miRNAs.

We then explored the significance of *cdh1* up-regulation in the *oct4* knockdown background (Fig 3D). For this, we targeted *cdh1* mRNAs using MOs against it in control MO or *oct4* MO-electroporated background. Interestingly, knockdown of *cdh1* caused a robust decline in MGPCs proliferation in a dose-dependent manner both in control and *oct4* MO-electroporated conditions (Fig 4I and J). The double knockdown of *cdh1* and *oct4* had a more dramatic decrease in MGPC number (Fig 4I and J), suggesting that these two genes may also have an independent influence on total MGPC number in regenerating retina. Based on these results, we could assume that Oct4 influences retinal MGPC number through *snail*, *zeb* family members, *miR-200a*/*miR-200b*/*miR143*/*miR-145*, and *cdh1*.

### Effects of Oct4 in vivo overexpression on gene regulation and cell proliferation

We next decided to see the influence of Oct4 overexpression in the zebrafish retina. For this, the *gfp-oct4* mRNA was transfected into the injured retina and was compared with *gfp* mRNA-transfected

control for the regenerative response. We followed an experimental timeline up to 2 dpi/4 dpi post *oct4* transfection (Fig 5A). Interestingly, the overexpression of Oct4 caused a decline in BrdU⁺ cells in the 4 dpi retina (Fig 5B and C). These results made us speculate that Oct4 if overexpressed, had a negative influence on cell proliferation. We then decided to use a range of *oct4* mRNA concentrations such as 150, 350, 750, and 1,000 ng to transfect into the injured retina. To our surprise, we found that there is an increase in cell proliferation in 150 and 350 ng that decreased thereafter in 750 and 1,000 ng (Figs 5D and E, and S6A). Furthermore, on closer evaluation, it is seen that the Oct4⁺ cells, after *oct4* mRNA transfection, were always present adjacent to the little number of BrdU⁺ cells (Fig 5D). At lower concentrations, Oct4 had a pro-proliferative effect, whereas at higher concentrations, it had an anti-proliferative effect. This probably could be because of differential partner selection by Oct4 at various concentrations in the injured retina. We speculated a differential affinity collaboration of Oct4 with transcriptional repressors such as histone deacetylase1 (Hdac1) during retina regeneration. Earlier studies have demonstrated the dependence of retina regeneration on differential regulation of Hdac1 in zebrafish (Mitra et al, 2018) and mice (Jorstad et al, 2017). To decipher if the differential collaboration of Oct4-Hdac1 existed, we decided to do a Co-IP of Oct4–Hdac1 complex using antibodies targeting both these proteins at various times postretinal injury. Interestingly, we found that Oct4 had less affinity to Hdac1 at early time points of retinal injury, which progressively enhanced towards the end of regeneration (Fig 5F). Based on these observations, we predicted that the differential effect of Oct4 overexpression on cell proliferation could also be under the influence of varying affinity to its collaborators such as Hdac1, which, as a transcriptional repressor, plays important roles during zebrafish retina regeneration. We then analyzed if the Hdac1 levels influenced the collaborative affinity of Oct4 at the early stages of retina regeneration. For this, we transfected injured retina with *hdac1* mRNA along with control *gfp* mRNA at 15 hpi. Interestingly, in *gfp* mRNA-transfected retina, we saw a similar affinity of Oct4 for Hdac1 (Fig 5G), while the overexpression of *hdac1* abolished the existing affinity at 15 hpi (Fig 5F and G). In addition, we also explored if the loss of Oct4–Hdac1 collaboration in Hdac1-overexpressed conditions caused an effect on the number of MGPCs in regenerating retina. Overexpression of Hdac1 significantly increased the number of MGPCs at 2 dpi (Fig S4C and D), suggesting that the Oct4–Hdac1 collaboration is necessary to keep the number of MGPCs within the desired limits at the site of injury.

Gene expression analysis in *oct4*-overexpressed retina showed that the levels of *zeb1a*/*zeb2a*, *miR-200a*/*miR-200b*, and *miR-143*/*miR-145* family genes were opposite to what we found in *oct4* knockdown background (Fig S4E and F). As discussed earlier, we

---

Oct4-BS, which is confirmed to be functional in a ChIP assay (lower) in the retina at 16 hpi. **(H)** qRT-PCR analysis of *cdh1* and *oct4* mRNA levels in the *zeb1a*-transfected retina at 2 dpi and 16 hpi, compared with *gfp* control. **(I)** The qRT-PCR analysis of *miR-200a*, *miR-200b*, *miR-143*, and *miR-145* genes in *oct4* knockdown retina at 2 dpi; *$P < 0.02$ (*t* test), N = 4. **(J, K)** The promoter schematics of *miR-200* family (gene cluster) (J, upper) and *miR-143*/*miR-145* (gene cluster) (K, upper) reveal the presence of Oct4-BSs, which are confirmed to be functional using antibodies against Oct4 (lower left), and Hdac1 (lower right) in a ChIP assay, at 16 hpi. **(L, M)** Western blot analysis of Co-IP of Hdac1 and Oct4 in retinal extracts at 16 hpi probed with anti-Hdac1 (L) and anti-Oct4 (M) antibodies. Ctl MO is control MO. Error bars are SD.

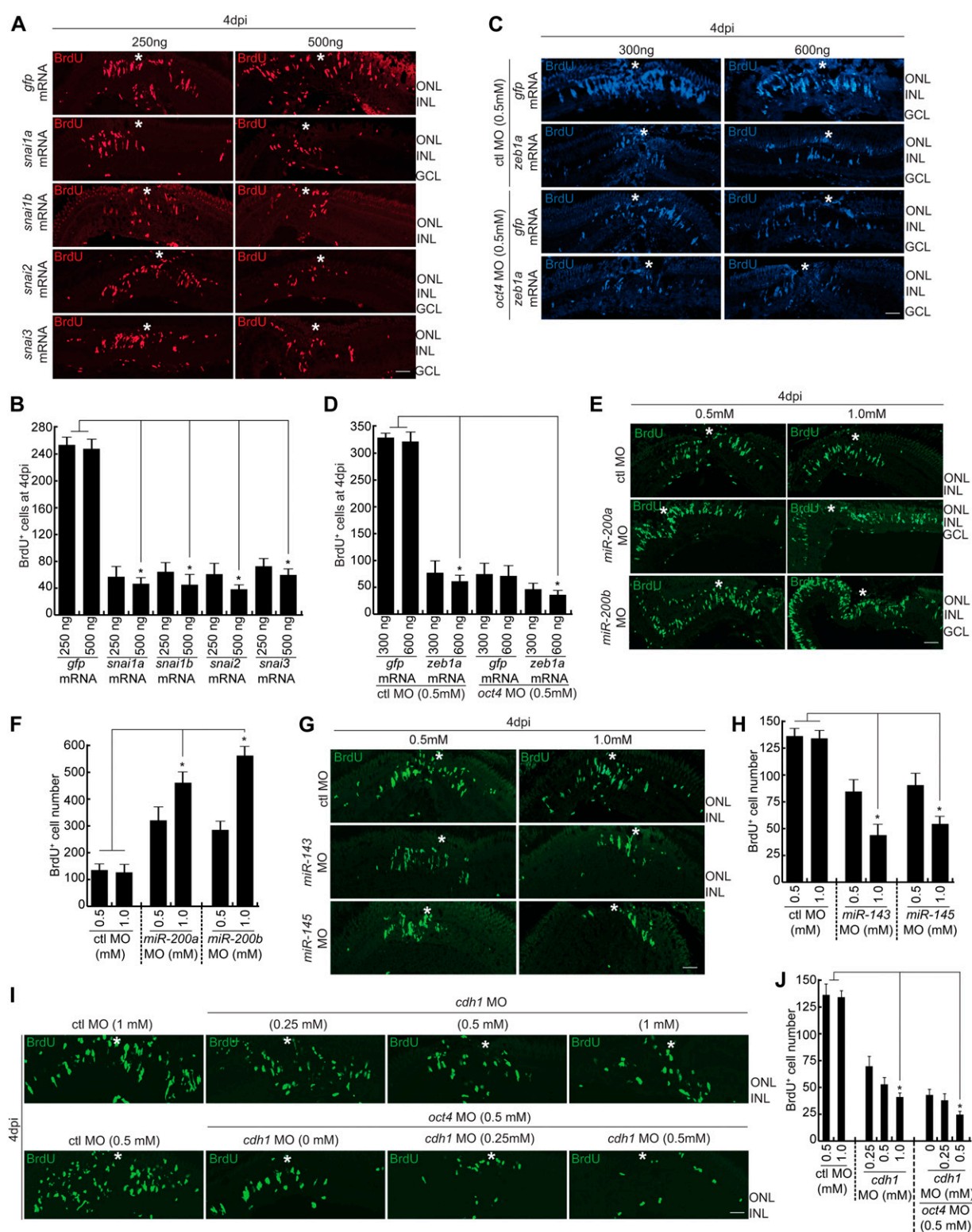

**Figure 4. Involvement of Snails, Zebs, *miR200a/miR-200b/miR-143/miR-145*, and Cdh1 to ensure adequate number of MGPCs during retina regeneration.**
**(A, B)** IF confocal microscopy images of 4 dpi retinal cross sections show BrdU⁺ MGPCs in *snai1a*, *snai1b*, *snai2*, and *snai3* mRNA-transfected conditions along with *gfp* mRNA-transfected control retina (A), which are quantified (B); *P < 0.002 (*t* test), N = 4. **(C, D)** IF confocal microscopy images of 4 dpi retinal cross sections show BrdU⁺ MGPCs in *zeb1a* mRNA transfected conditions along with *gfp* mRNA transfected control retina, with control MO and *oct4* MO-electroporated conditions (C), which are quantified (D); *P < 0.001 (*t* test), N = 4. **(E, F)** IF confocal microscopy images of 4 dpi retinal cross sections show BrdU⁺ MGPCs in *miR-200a/miR-200b* MO-electroporated conditions along with control MO (E), which are quantified (F); *P < 0.004 (*t* test), N = 4. **(G, H)** IF confocal microscopy images of 4 dpi retinal cross sections show BrdU⁺

found dual peaks of expression of Oct4 during retina regeneration in zebrafish, which includes a panretinal expression of *oct4* at 16 hpi (Fig 1A and B). Interestingly, the MO-mediated gene knockdown of *oct4* had an inhibitory effect on *zeb1a, zeb1b,* and *zeb2a*, whereas an up-regulation was seen with regards to *miR-200a* and *miR-200b* gene expressions at 2 dpi (Fig 3F and H). These results suggested that the initial panretinal Oct4 expression contributed to the MG reprogramming through the up-regulation of Zeb family members and down-regulation of *miR-200a/miR-200b/miR-143/miR-145* in the retina.

As we have already found increased levels of *snail* gene family members, namely, *snai1a, snai1b, snai2,* and *snai3* and Tgf-β signaling components, such as *tgfbr1b, tgfb2,* and its effector genes *tgfbi* and *smad7* in *oct4* knockdown background (Fig 3B and C), we also decided to examine the levels of these genes in *oct4*-overexpressed retina. We found an anticipated down-regulation of Tgf-β signaling components (Fig 5H) and *snail* genes (Fig 5I), which is suggestive of an Oct4-dependent Tgf-β signaling pathway that is active during retina regeneration. The double mRNA ISH of *oct4* and *tgfbi*, a Tgf-β signaling downstream gene, showed a relatively less frequent co-expression pattern in 4 dpi retina and a significant proportion of *oct4* expressing cells lacked *tgfbi* (Fig S4G). This observation also supported the view that Oct4 had a negative influence on Tgf-β signaling during retina regeneration.

Tgf-β signaling is known to suppress reprogramming (Li et al, 2010) and also suppress E-cadherin through Snails (Batlle et al, 2000). As we found a regulatory relationship of Oct4 with Tgf-β signaling, we explored if the *cdh1* levels also were affected in response to *oct4* overexpression. E-cadherin is a molecule that increases cellular proliferation (Park et al, 2017) and enables cellular adherence (van Roy & Berx, 2008), which is essential during MET (Wells et al, 2008). We saw a decline in *cdh1* mRNA levels in response to *oct4* overexpression in 2 dpi retina (Fig 5J). In agreement with the demonstrated decrease in MGPCs because of *cdh1* knockdown (Fig 4I and J), we presume that similar events could occur in Oct4-overexpressed retina wherein a dose-dependent down-regulation of *cdh1* is seen, along with lesser propensities of MG to switch into a proliferative phase.

Furthermore, the overexpression of Oct4 also caused a reduction in the expression of *lin28a* (Fig 5K) and an anticipated up-regulation in *let-7a* miRNA levels (Fig 5L) in 2 dpi retina. Interestingly, *oct4* mRNA transfection caused an up-regulation of *ascl1a* mRNA (Fig S4H and I) without causing a concomitant increase in its protein levels (Fig S4J), which is probably because of elevated levels of *let-7* miRNA, known to block the translation of *ascl1a* mRNA during retina regeneration (Ramachandran et al, 2010a; Mitra et al, 2018). These observations also support the idea that *let-7a* miRNA–mediated gene repression events would be crucial in contributing to the reduced MGPCs proliferation in Oct4 overexpressed retina.

## Oct4 is essential to bring an end to proliferative phase of MGPCs

Earlier observations showed a dual peak of expression of Oct4 in the injured retina and its differential collaboration with Hdac1 towards the late phase of regeneration. These observations prompted us to investigate whether Oct4 played alternative roles towards the end of the proliferative phase of retina regeneration. To explore this, we adopted a late gene knockdown approach at a time soon after the peak of proliferation seen at 4 dpi. We delivered *oct4*-targeting MO at the time of injury and electroporated later at 5 dpi and gave a BrdU pulse at 6 dpi. The retinae were harvested at 16 dpi as per the experimental timeline (Fig S5A). The rationale behind this experiment was to see if there is a continuation of active proliferation that occurs in the retina after *oct4* knockdown. Surprisingly, compared with the control MO-electroporated retina, we saw an increased number of BrdU-labeled cells (Fig S5B and C). These results could be due to two possible scenarios: (i) more MG cells enter the cell cycle and (ii) the BrdU⁺ MGPCs fail to exit the cell cycle and continue to be in the proliferative phase. To decipher which of these options prevailed in the late *oct4* knockdown retina, we adopted another experimental approach with an early BrdU and late EdU labeling of MGPCs. In an experimental timeline (Fig S5D), *oct4* MO was delivered at the time of injury followed by a BrdU pulse at 5 dpi, and electroporation after 3 h. The retinae were harvested at 8 dpi after 3 h of EdU pulse. Interestingly, compared with the control MO-electroporated retina, we saw a MO concentration-dependent increase in the number of EdU⁺ cells, which were also marked with BrdU in *oct4* knockdown retina at 8 dpi (Figs 6A–C, and S5D and E). We did not find a significant number of EdU⁺ cells that were not labeled with BrdU. These results supported the idea that the late knockdown of *oct4* from 5 to 8 dpi makes the MGPCs continue to proliferate. Early knockdown of *oct4* had an anti-proliferative effect with an associated increase in Tgf-β signaling component genes. Here, contrary to early inhibition of Oct4 (Fig 3B and C), its late knockdown regime (Fig 6A) had a negative influence on the Tgf-β signaling components (Fig 6D) and *snail* gene family members (Fig 6E), which also support the observed increase in MGPCs.

Moreover, we observed such a late knockdown of *oct4* was also associated with up-regulation of various cell cycle–specific genes such as *cyclins*, delta family members (Fig S5F), essential cytokines (Fig S5G) (Zhao et al, 2014; Mitra et al, 2018), and regeneration-associated transcription factors, namely, *ascl1a, mycb, oct4, sox2, lin28a,* and matrix metalloproteinases such as *mmp2* and *mmp9*. (Fig 6F). The *oct4* late knockdown had a profound effect on the down-regulation of *let-7a* miRNA levels (Fig 6G) that could be the effect of up-regulated *lin28a* (Figs 6F and S5H), which is known to facilitate MGPCs proliferation (Ramachandran et al, 2010a; Kaur et al, 2018). Similarly, the observed up-regulation of regeneration-associated transcription factors with late *oct4* knockdown at 8 dpi (Fig 6F) was also reflected in Western blot analysis (Fig 6H) and its quantification (Fig 6I). Based on these observations, we speculated

---

MGPCs in *miR-143/miR-145* MO-electroporated conditions along with control MO (G), which are quantified (H); *P < 0.01 (*t* test), N = 4. **(I, J)** IF confocal microscopy images of 4 dpi retinal cross sections show BrdU⁺ MGPCs in *cdh1* MO-electroporated conditions along with control MO and *oct4* MO (I), which are quantified (J); *P < 0.003 (*t* test), N = 4. Ctl MO is control MO. Error bars are SD. **(A, C, E, G, I)** Scale bars, 10 µm; the asterisk marks the injury site; GCL, ganglion cell layer; INL, inner nuclear layer; ONL, outer nuclear layer (A, C, E, G, I).

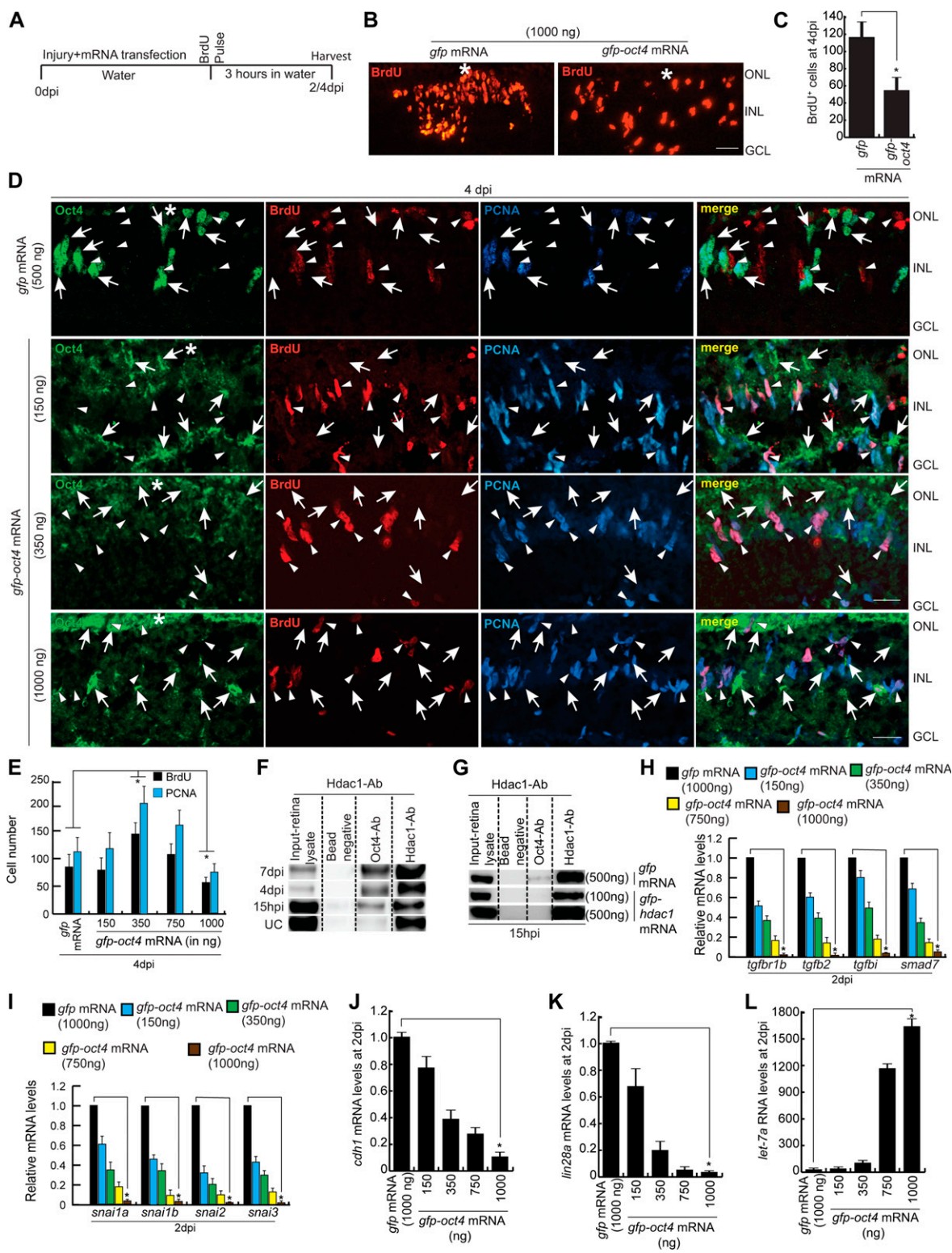

**Figure 5. Effect of Oct4 overexpression in the injured retina.**
**(A)** An experimental timeline that describes the mRNA transfection and BrdU pulse (for 4 dpi collection) before harvesting either at 2 or 4 dpi. **(B, C)** IF confocal microscopy images of retinal cross sections show reduced BrdU+ MGPCs at 4 dpi in *oct4* mRNA transfected condition, compared with *gfp* mRNA-transfected control retina (B), which is quantified (C); *P < 0.0001 (*t* test), N = 4. **(D)** IF confocal microscopy images of retinal cross sections of *oct4* mRNA-transfected retina at 4 dpi shows the cells with strong expression of Oct4 having a significant seclusion from PCNA+/BrdU+ MGPCs. White arrowheads mark BrdU+/PCNA+ cells and white arrows mark Oct4+ cells. **(E)** Quantification of BrdU+ and PCNA+ cells from *oct4*-overexpressed retina. **(F)** Western blot analysis of Co-IP of Oct4 and Hdac1 in retinal extracts at various time points postinjury probed with anti-Hdac1 antibody. **(G)** Western blot analysis of Co-IP of Oct4 and Hdac1 in retinal extracts obtained after *hdac1* overexpression at 15 hpi and

that Oct4 might be acting as a transcriptional repressor for several regeneration-associated genes during late phases of retina regeneration. These results also suggest that Oct4 may contribute to the regulation of component genes of repressive assembly such as NuRD complex.

### Oct4–NuRD interplay is essential for cell cycle exit of MGPCs

NuRD complex–mediated gene repressive events have been previously shown to be important for cellular differentiation (Hu & Wade, 2012). Here, because of late *oct4* knockdown, we found an increase in proliferation of MGPCs, which could result from a lack of NuRD complex–mediated gene repressive events. For this, we explored if the late *oct4* knockdown had any influence on the member genes of NuRD complex. Analysis of the gene family members encoding chromodomain helicase and DNA binding protein (Chd), namely, *chd3*, *chd4a*, and *chd4b*, along with *hdac1* that are members of NuRD complex showed a decline in late *oct4* knockdown retina (Fig 7A and B). This observation is supported by Oct4 binding onto the regulatory sequences of *hdac1* and *chd4a* genes, as revealed in ChIP assay (Fig 7C and D).

Furthermore, based on the above observations, we speculated that during the differentiation phase, Oct4 might alter its function from gene activation to repression through collaboration with Hdac1. To test this further, we adopted a late *hdac1* knockdown approach similar to the experimental timeline (Fig 7A). We found a similar increase in EdU⁺ MGPCs as found with late *oct4* knockdown, suggesting the involvement of Oct4–Hdac1 complex in gene repression events (Fig 7E and F). Notably, the late *hdac1* knockdown also caused a significant up-regulation of various regeneration-associated transcription factors, namely, *ascl1a*, *mycb*, *oct4*, *sox2*, *lin28a*, and matrix metalloproteinases such as *mmp2* and *mmp9* (Fig 7G), similar to what we found in late *oct4* knockdown. The regulatory DNA sequences of some of these transcription factors also had typical Oct4-BS. Furthermore, to explore if Hdac1 occupied these Oct4-BS, we performed a ChIP assay using Hdac1 antibody in the retinal extract from 6 dpi. Interestingly, Hdac1 occupied the Oct4-BS present on *ascl1a* (Fig 7H) and *oct4* (Fig 7I) gene promoters only in 6 dpi but not in 16 hpi retina (Fig 7H and I). Furthermore, overexpression of Hdac1 through mRNA transfection did not cause any change in its affinity for Oct4-BS on *ascl1a* and *oct4* promoters in 16 hpi retina (Fig 7H and I).

Notably, the late *hdac1* knockdown-mediated up-regulation of *lin28a* probably facilitated a decline in the *let-7a* miRNA levels (Fig 7J). The reduced levels of *let-7a* would also allow efficient translation and elevated protein levels of several regeneration-associated transcription factors (Fig 7K), also reflected by its quantification (Fig 7L). These results support the view that Oct4–Hdac1 complex contributes to the down-regulation of various regeneration-associated factors towards the late phases of regeneration. Such a regulation supports the view that the increased EdU⁺ cells seen in the late *oct4* knockdown retina are the result of

a genuine proliferative response of MGPCs. This type of differential influence of Oct4 at various stages of regeneration is probably mediated through its interaction with selective collaborating partners such as members of the NuRD complex.

We further explored if the increased number of MGPCs formed during the late *oct4* knockdown were able to differentiate into various retinal cell types. For this, a lineage tracing of these persistently proliferating MGPCs was performed by labeling with BrdU and their fate was followed up to 30 dpi. The cell type–specific staining and co-labeling with BrdU revealed that these MGPCs were indeed capable of differentiation (Fig S6B–D). We further confirmed these results by counting the number of BrdU-labeled cells that migrated to various retinal layers at 30 dpi (Fig 7M–O). These results support the significance of the Oct4-mediated gene regulatory network to cause cell cycle exit of MGPCs during retina regeneration.

## Discussion

Despite the knowledge on the expression patterns of PIFs soon after injury in the zebrafish retina, the roles played by Oct4 remained underexplored. In the present study, we delved into the significance of the induction of Oct4 soon after injury. We found differential roles played by Oct4 to cause a robust regenerative response in collaboration with a plethora of molecules, including transcription factors, components of Tgf-β signaling, miRNAs, and NuRD complex. Oct4 is one of the six PIFs that significantly influence several signaling pathways, which are necessary during cellular reprogramming in mammalian systems (Radzisheuskaya & Silva, 2014). As previously reported, the important regulatory network mediated by Oct4 during cellular reprogramming includes (i) repression of Tgf-β signaling (Li et al, 2010; Tan et al, 2015), (ii) activation of the *miR-200* family of miRNAs, which are the repressors of Zeb family of transcription factors (Wang et al, 2013), and (iii) epigenetic regulation of various genes responsible for cellular reprogramming in collaboration with Sox2 and Klf4 (Soufi et al, 2012; Buganim et al, 2013; Papp & Plath, 2013). Here, we explored if similar pathways ensue in the reprogramming of MG as a part of regeneration. The detailed findings from this study are summarized and depicted in a model (Fig 8).

We found a panretinal expression of *oct4* soon after retinal damage, which stays restricted to the site of injury throughout the proliferative cascade and secluded from the actively proliferating MGPCs. Closer analysis revealed that Oct4 induction during the proliferative phase of regeneration is a feature of MGPCs that quit the cell cycle. In spite of the report that Oct4 may not be important for somatic stem cell renewal (Lengner et al, 2007), studies in the injured mice retina that fails to regenerate showed an immediate induction of Oct4, which also declined quickly (Reyes-Aguirre & Lamas, 2016). However, in zebrafish, the dual expression peak of Oct4 at 16 hpi and 4 dpi in injured retina gave us critical clues about

---

probed with anti-Hdac1 antibody. **(H, I, J, K, L)** The qRT-PCR analysis reveals the levels of *tgfbr1b*, *tgfb2*, *tgfbi*, *smad7* (H), *snail*s (I), *cdh1* (J), *lin28a* (K), and *let-7a* miRNA (L) in *oct4* mRNA-transfected retina at 2 dpi; *P < 0.001 (*t* test), N = 4. Error bars are SD. **(B, D)** Scale bars, 10 μm; the asterisk marks the injury site; GCL, ganglion cell layer; INL, inner nuclear layer; ONL, outer nuclear layer (B, D).

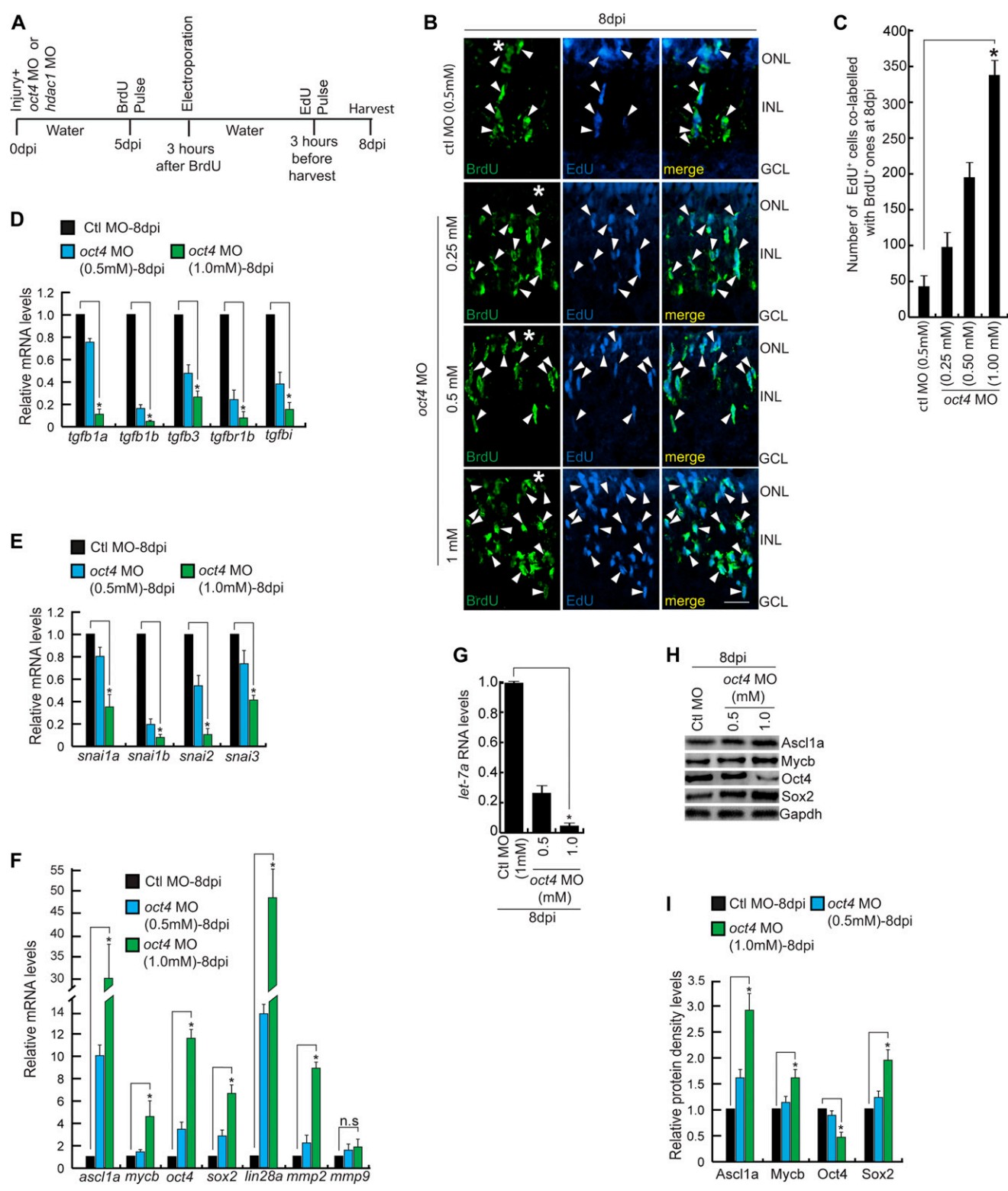

**Figure 6. The increased MGPCs seen in late *oct4* knockdown in regenerating retina is caused by the delay in cell cycle exit.**
**(A)** An experimental timeline that describes the injury, MO injection, BrdU pulse, late electroporation of the retina, and EdU pulse 3 h before harvest at 8 dpi. **(B, C)** IF confocal microscopy images of retinal cross sections show increased BrdU⁺ MGPCs at 8 dpi in *oct4* knockdown from fifth day onwards and a proof of the delay in quitting cell cycle revealed by EdU co-labeling with BrdU⁺ MGPCs (B), which is quantified (C); *P < 0.001, N = 4. White arrowheads mark BrdU⁺/EdU⁺ cells in (B). **(D)** qRT-PCR analysis of Tgf-β signaling component genes and its reporter *tgfbi* mRNA levels in late *oct4* knockdown retina, at 8 dpi; *P < 0.02 (*t* test), N = 4. **(E)** qRT-PCR analysis of *snail* family genes' mRNAs in late *oct4* knockdown retina, at 8 dpi; *P < 0.03 (*t* test), N = 4. **(F)** qRT-PCR analysis of *ascl1a*, *mycb*, *oct4*, *sox2*, *lin28a*, *mmp2*, and *mmp9* mRNA levels in late *oct4* knockdown retina, at 8 dpi. **(G)** qRT-PCR analysis of *let-7a* miRNA levels show a decline because of *oct4* late knockdown in 8 dpi retina. **(H, I)** Western blot analysis of

its importance and the possible existence of differential roles at early and late phases of regeneration. This led us to explore whether Oct4 performed similar or different roles during zebrafish retina regeneration in comparison with mammalian cellular reprogramming. We found the necessity of Oct4 during the early stages of retina regeneration to keep the Tgf-$\beta$ signaling at bay, which is similar to the mammalian system. Tgf-$\beta$ signaling is also known to be anti-proliferative during retina regeneration in various model organisms (Close et al, 2005; Lenkowski et al, 2013; Todd et al, 2017). In mammalian cellular reprogramming, Oct4 activates *miR-200* family (Radzisheuskaya & Silva, 2014), which is the repressor of *zeb* mRNAs (Park et al, 2008). Surprisingly in the zebrafish retina, we saw the direct binding of Oct4 on the promoters of *miR-200* and *zeb* family of genes resulting in repression of the former and activation of the latter. Moreover, considering the repressive role of *miR-200* on *zeb* mRNAs, one could presume that Oct4 ensures high levels of Zeb proteins in the early stages of retina regeneration. More importantly, the panretinal induction of Oct4 ensures elevated levels of *zeb*, probably to avoid reprogramming of MG away from the site of injury. Evidence for this is also seen in *1016tuba1a*:GFP transgenic retina wherein the GFP⁻ cells express higher levels of *zeb* than the GFP⁺ ones. Our overexpression studies of *zeb1a* and *zeb2a* in regenerating retina also confirmed their anti-proliferative nature as seen in cancer cells (Hugo et al, 2013). Similarly, the proliferating group of MGPCs also have cells that are about to differentiate and also the ones which would continue to be in the cell cycle for a prolonged time. Here too, the actively proliferating MGPCs have less Oct4 ensuring their persistence in the cycling phase, and the cells that are about to differentiate have relatively more Oct4 expression, which in turn activates pathways that block cell division or further reprogramming. This scenario is evident from the lack of proliferation in Zeb/Snail overexpressed retina.

Furthermore, it is interesting to note that Oct4 had a repressive role on *miR-200a*/*miR-200b* and *miR-143*/*miR-145*, which are anti-proliferative and pro-proliferative in nature, respectively. Although it may appear to be a conundrum, this type of transcriptional regulation of miRNAs by Oct4 ensures adequate MG reprogramming and induction of MGPCs, a situation otherwise could have led to undesirable cellular proliferation. In addition, we also showed that the Oct4 negatively influenced the E-cadherin in regenerating retina, unlike reported in the mammalian adult stem cells and during MET (Radzisheuskaya & Silva, 2014; An et al, 2017). The decreased *zeb* levels in early *oct4* knockdown conditions also could contribute to the elevated *cdh1* (E-cadherin) levels. The higher levels of Oct4 in post-proliferative MGPCs inevitably deemed them to adopt a cell differentiation cascade than a reprogramming one, justifying the observed results. These findings are also supported by the retinal Oct4 overexpression performed in this study. The MO-mediated knockdown of *cdh1* in isolation or in combination with *oct4* MO had a negative effect on MGPCs proliferation. It is also important to note that E-cadherin is known to promote neurite outgrowth from retinal ganglion cells (Oblander et al, 2007) and cellular proliferation in certain cancers (Dong et al, 2012). These observations suggest that an increase in *cdh1* levels in *oct4* knockdown retina was not able to complement the absence of Oct4 and the decline in the levels of *cdh1* alone was sufficient to hinder MGPCs induction.

We discovered that the significance of Oct4 during retina regeneration seemed to be multifactorial at various phases of retina regeneration, which is revealed from our early and late knockdown experiments of Oct4. In other words, the early *oct4* knockdown soon after the injury had an anti-proliferative effect, whereas the late one had a pro-proliferative effect on MGPCs. Furthermore, the overexpression of Oct4 through mRNA transfection of the retina had a negative effect on proliferation, especially with higher concentration. Also, the overexpression of *hdac1*, one of the collaborators of Oct4, abolished the Hdac1–Oct4 interaction and this also accelerated the MGPC proliferation in the retina, suggesting the necessity of selective extent of collaboration of Oct4 with its partners in regulating a particular gene target. It is also important to note that zebrafish Oct4 failed to support murine embryonic stem cell self-renewal (Morrison & Brickman, 2006) probably because of its differential effects in comparison with mammalian counterpart. This opened up the scenario for us to explore the intricate details about the components of the Oct4-mediated regulatory network at different phases of retina regeneration.

Our study showed that Oct4 played differential roles in regulating the proliferation of MGPCs at various phases of retina regeneration. Closer analysis revealed that the MGPCs that are formed during regeneration failed to quit the cell cycle and continue to be in the proliferative phase in late *oct4* knockdown. At later stages, Oct4 switches its activating function on genes that facilitate cell proliferation to their suppression in collaboration with Hdac1, a member of NuRD complex. Oct4 positively regulated the transcriptional repressor Her4.1 to keep *lin28a* at bay in cells that exit cell cycle. Furthermore, the up-regulation of *lin28a* and a decline in the *let-7a* miRNA levels in late *oct4* knockdown could also facilitate the translation of regeneration-associated factors as previously demonstrated (Ramachandran et al, 2010a; Kaur et al, 2018). In conclusion, our results illustrated various mechanisms of retina regeneration mediated through the PIF, Oct4. This study also opens up new vistas of exploration in similar lines that would enable designing therapeutic strategies to cure mammalian retinal blindness.

## Materials and Methods

### Animals

Zebrafish were maintained at 26°C–28°C on a 14:10 h light/dark cycle. The *1016tuba1a*:GFP transgenic fish used in this study have been previously characterized (Fausett & Goldman, 2006). Embryos for microinjection in Luciferase assays were obtained by natural breeding of wild-type fish.

---

various regeneration-associated factors in late *oct4* knockdown retina at 8 dpi, which is quantified by densitometry (I). Gapdh is used as the loading control. Ctl MO is control MO. Error bars are SD. **(B)** Scale bars, 10 *μm*; the asterisk marks the injury site; GCL, ganglion cell layer; INL, inner nuclear layer; ONL, outer nuclear layer (B).

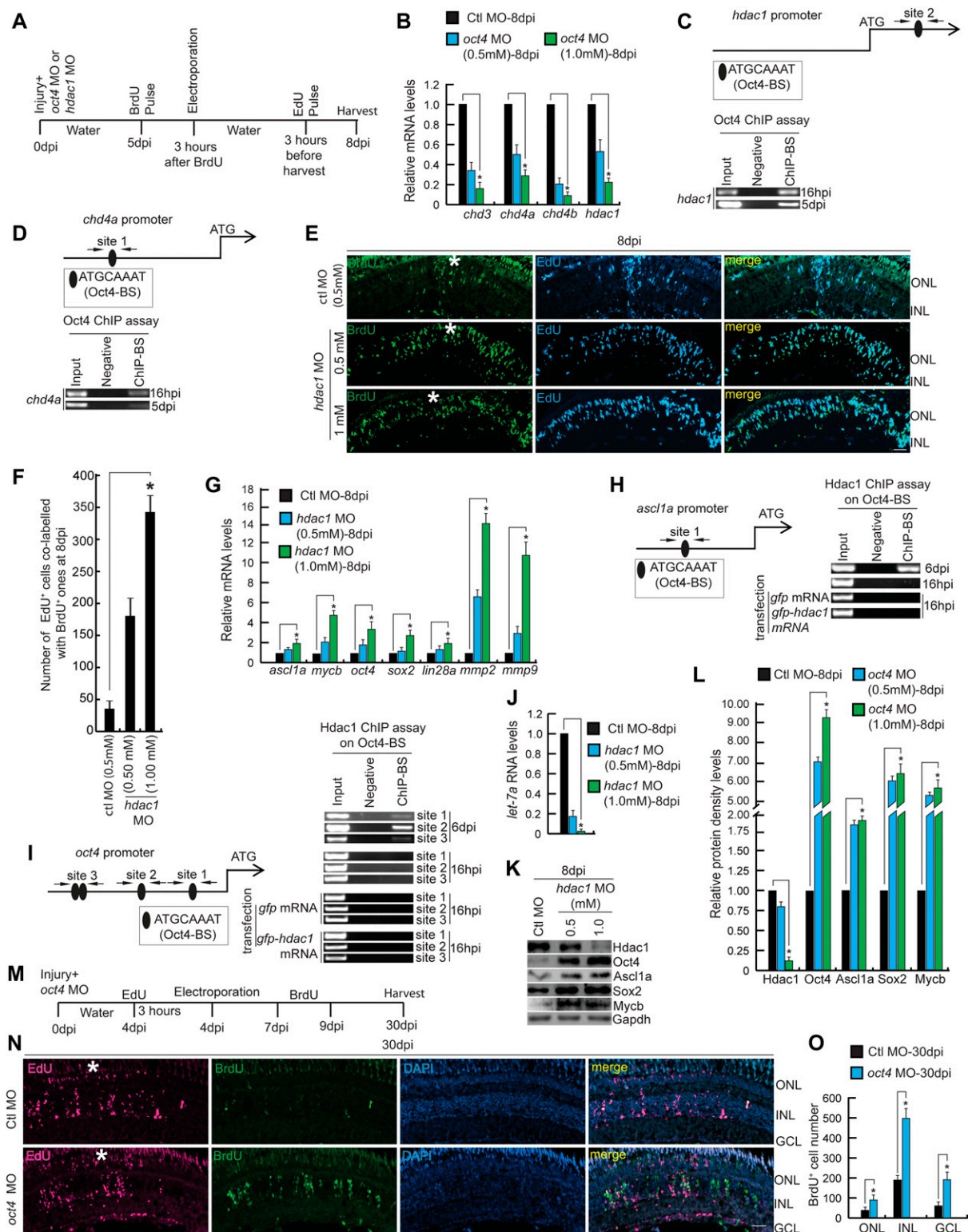

**Figure 7. Effect of late *hdac1* and *oct4* knockdowns on MGPCs and gene expressions.**
**(A)** An experimental timeline that describes the injury, MO injection, BrdU pulse, late electroporation of the retina, and EdU pulse 3 h before harvest at 8 dpi. **(B)** qRT-PCR analysis of NuRD complex component genes' mRNA levels in late *oct4* knockdown retina, at 8 dpi. **(C, D)** The *hdac1* (C) and *chd4a* (D) promoter schematics reveal the typical Oct4-BSs (upper) and the retinal ChIP assays confirm the physical binding of Oct4 at these sites (lower), in 16 hpi and 5 dpi retina. **(E, F)** IF confocal microscopy images of retinal cross sections show increased BrdU⁺ MGPCs at 8 dpi in *hdac1* knockdown from fifth day onwards and the delay in exiting cell cycle as revealed by EdU co-labeling with BrdU⁺ MGPCs (E), which is quantified (F). **(G)** qRT-PCR analysis of *ascl1a, mycb, oct4, sox2, lin28a, mmp2,* and *mmp9* mRNA levels in late *hdac1* knockdown

## Anesthesia and retinal injury

Fish were anesthetized with tricaine methanesulfonate. The retinal injury was performed with a 30G needle as described previously (Fausett & Goldman, 2006). All experiments were performed to a minimum of four times for consistency and s.d.

## Primers and plasmid construction

All primers are listed in Table S1. The *ascl1a*:GFP–luciferase construct was described previously (Ramachandran et al, 2010a; Wan et al, 2012). Full coding sequence (CDS) of *oct4*, *hdac1*, *snai1a*, *snai1b*, *snai2*, *snai3*, *zeb1a*, and *zeb2a* were cloned by PCR amplification of cDNA prepared from RNA of 24 h postfertilization zebrafish embryos, using their respective primer pairs. Postdigested PCR amplicons of *oct4* and *hdac1* were cloned in pCS2⁺-GFP, whereas the *zeb1a*, *zeb2a*, and *snail* were cloned in pCS2⁺. The pCS2⁺-GFP plasmid was described earlier (Mitra et al, 2018, 2019).

## mRNA synthesis and embryo microinjection

Gene clones of *oct4*, *hdac1*, *snai1a*, *snai1b*, *snai2*, *snai3*, *zeb1a*, and *zeb2a*-CDS were linearized and capped mRNAs were synthesized using the mMESSAGE mMACHINE SP6 (AM1340; Thermo Fisher Scientific) in vitro transcription system. For luciferase assays, single-cell zebrafish embryos were injected with a total volume of ~1 nl solution containing 0.02 pg of *Renilla* luciferase mRNA (normalization), 5 pg of *promoter*:GFP–luciferase vector and 0–6 pg of *oct4* mRNA or 0.1–0.5 mM *oct4* MO. To assure consistency of results, a master mix was made for daily injections and ~300 embryos were injected at the single cell stage. After 24 h, the embryos were divided into three groups (~70 embryos/group) and lysed for dual-luciferase reporter assays (E1910; Promega).

## mRNA transfection

mRNA transfection was performed for in vivo overexpression of *oct4*, *hdac1*, *snai1a*, *snai1b*, *snai2*, *snai3*, *zeb1a*, and *zeb2a*, in injured zebrafish retina. Transfection mixture contained two solutions constituted in equal volumes: (i) 4–5 μg of mRNA mixed with HBSS, (ii) Lipofectamine messenger max reagent (cat. no. LMRNA001; Invitrogen) mixed with HBSS. Both the solutions were allowed to stand at room temperature for 10 min and then mixed dropwise followed by 30-min incubation at room temperature. The resultant solution was used for injection in zebrafish retina followed by electroporation as described earlier (Fausett & Goldman, 2006). The *gfp* mRNA transfection was performed in control injured retina and in retinae transfected with *snails* and *zebs*-mRNA.

## MO electroporation and knockdown rescue

Lissamine-tagged MOs (Gene Tools) of ~0.5 μl (0.25–1.0 mM) volume were injected, at the time of injury, using a Hamilton syringe of 10 μl volume capacity. MO delivery to cells was accomplished by electroporation with five pulses at 70 V for 50 ms with a gap period of 950 ms in between the pulses (Fausett et al, 2008). The fish retinae were assayed for cell death post electroporation and compared with control before proceeding with actual experiments. We did not observe any cell death because of electroporation to the retina. The sequence of control MO has been previously described (Wan et al, 2012). MOs targeting *oct4*, miRNAs, and *cdh1* are as follows:

> *oct4* MO targeting 5′UTR, 5′-CTTTCCGCTAAAAAGGTTGTTGAGA-3′
> 2-*oct4* MO, 5′-GCTCTCTCCGTCATCTTTCCGCTAA-3′
> *miR-200a* MO, 5′-ACATCGTTACCAGACAGTGTTA-3′ (Flynt et al, 2009)
> *miR-200b* MO, 5′-TCATCATTACCAGGCAGTATTA-3′ (Flynt et al, 2009)
> *miR-143* MO, 5′-GAGCTACAGTGCTTCATCTCA-3′ (Lagendijk et al, 2011)
> *miR-145* MO, 5′-GGGATTCCTGGGAAAACTGGAC-3′ (Lagendijk et al, 2011)
> *cdh1* MO, 5′-ATCCCACAGTTGTTACACAAGCCAT-3′ (Xiong et al, 2014).

In vivo rescue experiments were designed for testing the specificity of *oct4* antisense oligos. We did the transfection of zebrafish retina using *oct4*-specific mRNA alongside the MO targeting 5′ UTR region. For confirming the efficient mRNA transfection, GFP mRNA was also delivered by transfection in control retina along with either *oct4* MO or control MO, whereas GFP fusion with *oct4* mRNA was used in other sets.

## Total RNA isolation, RT-PCR, and qRT-PCR analysis

Total RNA was isolated from dark-adapted zebrafish retinae of control, injured, and drug-treated/MO-electroporated/mRNA-transfected groups using TRIzol (Invitrogen). A combination of oligo-dT and random hexamers were used to reverse-transcribe ~5 μg of RNA using Superscript III Reverse Transcriptase (Invitrogen) to generate cDNA. PCR reactions used Taq or Phusion (New England Biolabs) polymerase and gene-specific primers (Table S1) with previously described cycling conditions (Ramachandran et al, 2010a). Quantitative real-time PCR (qRT-PCR) was carried out in triplicate with KOD SYBR (SYBR green containing PCR mix with KOD DNA polymerase from *Thermococcus kodakaraensis*) qRT-PCR mix (QKD-201; Genetix) on a real-time PCR detection system (MasterCycler RealPlex4; Eppendorf). The *let-7a miRNA* levels were determined with TaqMan *hsa-let7-a* probe (Applied Biosystems) as

---

retina, at 8 dpi. **(H, I)** The *ascl1a* (H) and *oct4* (I) promoter schematics reveal the typical Oct4-BS (upper) and the retinal ChIP assays confirm the physical binding of Hdac1 at the Oct4-BS (lower) in 6 dpi retina. The ChIP assay performed in 16 hpi retina and also in *hdac1*-overexpressed condition reveal no binding of Hdac1 at Oct4-BS of *ascl1a* (H, right) and *oct4* (I, right) promoters. The *gfp* mRNA transfection is the control. **(J)** The qRT-PCR analysis shows decreased *let-7a* miRNA levels with late *hdac1* knockdown at 8 dpi. **(K, L)** Western blot analysis of different regeneration-associated factors in late *hdac1* knockdown retina at 8 dpi, which is quantified by densitometry (L). Gapdh is used as the loading control. **(M)** An experimental timeline that describes the injury, MO injection, EdU pulse, and late electroporation of the retina at 4 dpi and BrdU on 7–9 dpi before harvest at 30 dpi. **(N, O)** IF confocal microscopy images of retinal cross sections show EdU and BrdU-labeled MGPCs in *oct4* knockdown from the fourth day onwards and the localization of the BrdU-labeled MGPCs to various retinal layers at 30 dpi (N), which is quantified (O). Ctl MO is control MO. Error bars are SD. **(E, N)** Scale bars, 10 μm; the asterisk marks the injury site; GCL, ganglion cell layer; INL, inner nuclear layer; ONL, outer nuclear layer (E, N).

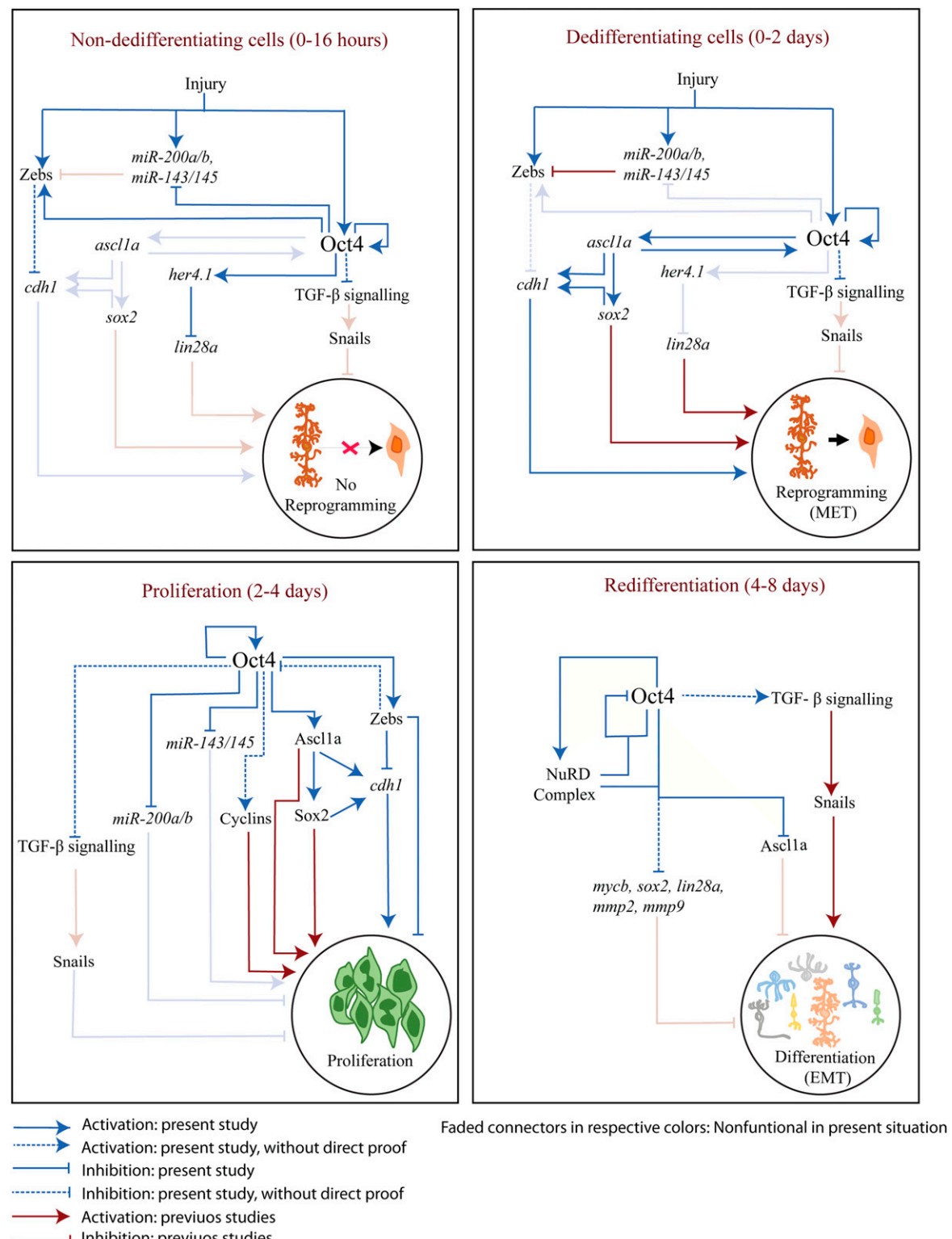

**Figure 8. The gene regulatory network mediated through Oct4 regulatory axes in different phases of retina regeneration.**
The model schematically describes gene regulatory mechanisms of various regeneration-associated factors discovered in this study along with already reported ones, at different stages of retina regeneration.

per the manufacturer's instructions. The relative expression of mRNAs in control and injured retinae was deciphered using the ΔΔCt method and normalized to *β-actin* mRNA levels.

### Co-IP and ChIP assay

Co-IP and ChIP assays were carried out in adult retina at different time points using ~20 adult retinae after dark adaptation. Chromatin was isolated as described previously (Lindeman et al, 2009). After sonication, a part of chromatin was kept as input and remaining was distributed into two equal aliquots; one of them was pulled down with anti-Oct4, anti-Hdac1, anti-Ascl1a, or anti-Sox2 antibodies separately (described below) and other half was pulled down with rabbit IgG (Sigma-Aldrich) as negative control. Primers used for ChIP assays are described in Table S1.

For Co-IP, retinae were frozen at –80°C in lysis buffer containing protease inhibitor cocktail and PMSF. Retinal lysate was prepared by thawing the sample in water followed by pipetting and vortexing. Lysate mixed with fresh lysis buffer containing protease inhibitor and PMSF was centrifuged at 9,425*g* for 10 min at 4°C. Supernatant separated from cell debris was subjected to pull down with anti-Oct4 and anti-Hdac1 antibodies. Co-immunoprecipitated sample was lysed in Laemmli buffer and subjected to Western blotting as described below.

### Western blotting and quantification

Western blotting was performed using six retinae per experimental sample, lysed in Laemmli buffer, size-fractioned in 12% acrylamide gel at denaturing conditions, and transferred onto Immuno-Blot polyvinylidene fluoride (PVDF) membrane (cat. no. 162-0177; Bio-Rad), followed by probing with specific primary antibodies and HRP-conjugated secondary antibodies for chemiluminescence assay using Clarity Western ECL (cat. no. 170-5061; Bio-Rad). Western blotting images were quantified with ImageJ software and the values obtained were normalized to loading control (Gapdh) in each experimental setup. Fold change in protein expression of different samples was determined in comparison with the control injured retinae.

### BrdU/EdU labeling, retina tissue preparation, immunofluorescence, ISH, FISH, and antibodies used

BrdU labeling was performed by a single intraperitoneal injection of 20 *μ*l of BrdU (20 mM) 3 h before euthanasia and retina dissection unless mentioned specifically. EdU labeling was done by intravitreal injection of 10 mM EdU solution as described earlier (Mitra et al, 2018, 2019). Fish were given a higher dose of tricaine methanesulphonate and eyes were dissected, lens removed, fixed in 4% paraformaldehyde, and sectioned as described previously (Fausett & Goldman, 2006). The mRNA ISH was performed on retinal sections with fluorescein (FL) or digoxigenin (DIG)-labeled complementary RNA probes (FL/DIG RNA labeling kit; Roche Diagnostics) (Barthel & Raymond, 2000). FISH was performed according to the manufacturer's directions (cat. no. T20917, B40955, and B40953; Thermo Fisher Scientific). Sense probes were used in every ISH separately as control, to assess the potential of background signal. Immunofluorescence microscopy protocols and antibodies were previously

described (Ramachandran et al, 2010b; Wan et al, 2012; Mitra et al, 2019).

Other primary antibodies used for Western blotting and immunofluorescence were rabbit polyclonal antibody against Oct4 (AB3209; Merck), mouse monoclonal against Oct3/4 (sc5279; Santa Cruz Biotechnology), mouse polyclonal antibody against GFP (cat. no. ab38689; Abcam), rabbit polyclonal against GFP (cat. no. ab290; Abcam), rabbit polyclonal antibody against Sox2 (cat. no. ab59776; Abcam), rabbit polyclonal antibody against Hdac1 (cat. no. ab41407; Abcam), and rabbit polyclonal antibody against GAPDH (cat. no. SAB2701826; Sigma-Aldrich). Secondary antibodies used were goat anti rat/mouse/rabbit tagged to fluorescent dyes ranging from Alexa Fluor 488–647. The secondary antibody used in Western blotting analysis was HRP-conjugated anti-rabbit antibody.

### Fluorescence and confocal microscopy, cell counting, and statistical analysis

After the completion of staining experiments, the slides were examined with a Nikon N*i*-E fluorescence microscope equipped with fluorescence optics and Nikon A1 confocal imaging system. The PCNA[+] and BrdU[+] cells were counted by observation of their fluorescence in retinal sections. ISH[+] cells were visualized through differential interference contrast in the same microscope and quantified. Observed data were analyzed for statistical significance by comparisons done using a two-tailed unpaired *t* test to analyze data from all experiments. Error bars represent SD in all histograms.

### Fluorescence-based cell sorting

RNA was isolated from FACS-purified MG and MG-derived progenitors at 4 dpi as previously described (Ramachandran et al, 2011, 2012b). Briefly, uninjured and injured retinae were isolated from *1016tuba1a*:GFP transgenic fish. GFP[+] MGPCs from *1016tuba1a*:GFP retinae at 4 dpi were isolated by treating retinae with hyaluronidase and trypsin and then sorted on a BD FACS Aria Fusion high-speed cell sorter. Approximately 30 injured retinae with 10 pokes per retina from *1016tuba1a*:GFP fish yielded 70,000 GFP[+] and 150,000 GFP[−] cells.

## Supplementary Information

## Acknowledgements

P Sharma acknowledges postdoctoral fellowship support from the Wellcome Trust/DBT (Department of Biotechnology) India Alliance and Indian Institute of Science Education and Research (IISER), Mohali. S Gupta acknowledges her support from the Indian Council of Medical Research for Senior Research Fellowship. M Chaudhary, S Mitra, MA Khursheed, and B Chawla acknowledge their financial support from the IISER, Mohali. This work was supported by the Wellcome Trust/DBT India Alliance Intermediate Fellowship (IA/I/12/2/

500630) awarded to R Ramachandran. R Ramachandran also acknowledges research funding from Science Education and Research Board, Department of Science and Technology, India (EMR/2017/001816), DBT, India (BT/PR9407/BRB/10/12612013), (BT/PR17912/MED/31/336/2016), and support from IISER Mohali.

## Author Contributions

P Sharma: resources, data curation, formal analysis, validation, investigation, visualization, methodology, and writing—review and editing.
S Gupta: resources, methodology, and writing—review, and editing.
M Choudhary: resources, methodology, and writing—review, and editing.
S Mitra: resources.
B Chawla: resources.
MA Khursheed: resources and methodology.
R Ramachandran: conceptualization, resources, data curation, software, formal analysis, supervision, funding acquisition, validation, investigation, visualization, project administration, and writing—original draft, review, and editing.

## Conflict of Interest Statement

The authors declare that they have no conflict of interest.

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
