## [Reviewer comments · Life Science Alliance]

Life Science Alliance

Oct4 mediates Müller glia reprogramming and cell cycle-exit during retina regeneration in zebrafish

Poonam Sharma, Shivangi Gupta, Mansi Choudhary, Soumitra Mitra, Bindia Chawla, Mohammad Khursheed, and Rajesh Ramachandran

DOI: <https://doi.org/10.26508/lsa.201900548>

Corresponding author(s): Rajesh Ramachandran, Indian Institute of Science Education and Research, Mohali

Review Timeline:

Submission Date:	2019-09-11
Editorial Decision:	2019-09-12
Revision Received:	2019-09-13
Editorial Decision:	2019-09-17
Revision Received:	2019-09-24
Accepted:	2019-09-24

Scientific Editor: Andrea Leibfried

Transaction Report:

Please note that the manuscript was previously reviewed at another journal and the reports were taken into account in the decision-making process at Life Science Alliance.

Referee #2 Review

Report for Author:

A previous major concern of mine was that the proposed mechanism downstream of Oct4 (which included Zeb1/2, Ecad, Snails and miRNAs) was not sufficiently supported by the data. In this review, the authors validated that these downstream factors are indeed activated in MG, that they influence MG proliferation, and they change with Oct4 manipulations. This is a significant improvement.

The new antibody for Oct4 is also much better and the quality of immunolabeling has been improved in this version.

I do not have any qualms with the individual experiments. However, the overall mechanisms proposed regarding Oct4 in regeneration leave the reader very confused. Oct4 is proposed to regulate MGPC formation by (1) suppressing Snail factors to allow proliferation, (2) driving Zeb1 to not proliferate too much, (3) controlling four different miRNAs, half of which promote proliferation, the other half which suppress proliferation, (4) suppressing Cdh1 (low Cdh1 levels further suppress proliferation, yet when Oct4 is knocked down increased Cdh1 levels did not increase proliferation). Furthermore, whether Oct4 is promoting cell cycle re-entry or exit is dependent on concentration and partnership with Hdac1. Maybe this is just the nature of the data, but it leaves the reader very confused and the focus of the manuscript seems scattered and jumps all over the place. If this work is relevant to others investigating retinal regeneration I am not sure what experiments regarding Oct4 this data would suggest.

Referee #3 Review

Report for Author:

The authors have made significant effort to improve the manuscript. The quality of images and photomicrographs is much better than in the prior iteration. Most of the comments of the reviewers appear to be adequately addressed.

I appreciate the effort to demonstrate no cell death or proliferation of MGPCs with electroporation, however the defensive response of the authors is not appreciated. Nearly all papers from Hyde, Thummel and Hitchcock that make use of in vivo electroporation of zebrafish retina mention damage incurred by the technique and exclusion of individuals where excessive retinal damage was caused by the electroporation. It is commendable that the authors incur no cell death or damage in the retina at 4 days after treatment, but this does not exclude the possibility of damage and death incurred at earlier times following electroporation. In any event, consideration of electroporation-mediated cell damage/death (particularly when combined with MO's which can also be cytotoxic) shouldn't be entirely dismissed and ignored from interpretations.

Referee #1 Review

Report for Author:

This excellent piece of work shows that Octamer-binding transcription factor 4 (Oct4) is an essential for reprogramming of Müller cells into a proliferating population of progenitors. The paper highlights that several regeneration-associated factors (including Ascl1a, Lin28a, Sox2, Zeb, E-cadherin, and various microRNAs) are targets for Oct4 regulation.

The experiments were carried out with state-of-the-art technologies and the manuscript is well written.

I have no major points of criticism, which is a very rare occasion in my experience as a reviewer. As a minor point, the term Müller cell should be written as "Müller" and not "Muller". The cell is named after the German scientist Heinrich Müller, who discovered the cells in the 1840s.

Referee #2 Review

Report for Author:

This work by Sharma, et al. implicates the pluripotency gene Oct4 in retinal regeneration in the zebrafish model. In this model, Muller glia cells respond to a stab injury by reprogramming into Muller glia-derived progenitor cells (MGPCs) that can regenerate retinal neurons. The authors posit that Oct4 plays a critical and diverse role in different phases of regeneration. In early phases after injury, the authors convincingly demonstrate that Oct4 is necessary for MGPC induction and that this probably occurs through modulation of various regeneration associated genes as well as an *ascl1a* regulatory loop. At later time points the authors suggest Oct4 switches to a transcriptional repressor by associating with Hdac1 and NurD complex and this switch now controls cell cycle exit and repression of regeneration associated genes.

The function of Oct4 in retinal regeneration has yet to be explored in the field of retinal regeneration and the authors suggest a novel and interesting biphasic mechanism. However, many conclusions drawn do not follow from the data shown and further experiments are needed to provide insight into the mechanisms downstream of Oct4.

Major point 1. (Figs 3 and 4) The proposed mechanism for how Oct4 controls MGPC induction involves downregulation of TGF β components, activation of Zeb1/2 and E-cadherin and regulation of miR-200a/b and mir143/145. However, this is not sufficiently shown. At numerous points throughout the manuscript language is used such as "it is logical to believe that Oct4 expression indeed upregulates Zeb genes expression to prevent further reprogramming (line 290)". The authors do provide evidence that these above listed factors are induced after injury and contain Oct4 binding sites occupied by Oct4. Does manipulation of Zeb1/2, E-cad, Snails, or the miR's mentioned actually impact regeneration? As it stands these are correlative phenomenon (mRNA is induced and binding sites are occupied) and do not imply that these factors control MGPC formation or proliferation.

Major point 2: The authors propose an interesting biphasic effect where Oct4 promotes MGPC formation early after injury while at later time points it represses reprogramming genes to induce cell cycle exit. More data would be needed to make this a compelling case. The first set of data provided that Oct4 has opposing effects is not based on "regenerative time" but rather on overexpressing Oct4. The authors show when Oct4 is overexpressed (only at high levels) it switches to a repressive function. They link this to "regenerative time" by doing a co-IP with the repressive factor Hdac1 and find that these two factors form a complex at later time points in regeneration but not earlier. This story between Oct4 levels and phase of regeneration is confusing. Oct4 expression naturally peaks at earlier time points after injury, yet that's when it has the least affinity to Hdac1. So in the normal regenerative context the affinity of Oct4 to Hdac1 does not seem to be related to the levels of Oct4. So why do the levels of Oct4 matter when overexpressed at later time points? Does Oct4 switching from an activator to a repressor depend on the levels of Hdac1 or NurD? And if so, if you overexpressed HDAC1 early after injury would you then prematurely switch Oct4 from an activator of regeneration genes to a repressor?

Other concerns.

3. Figure 1c. The authors state that Oct4 is first induced panretinally and then restricted to the injury site at later stages post injury. The images for this conclusion are unclear at the later stages (4dpi and 7dpi) the ISH signal is present across the whole field of view presented. Images with a wider field of view would be more convincing.

4. Figure 1f. The pattern of staining for the Oct4 antibody is unconvincing. I do believe that Oct4 is in MGPCs (the signal in tuba1aGFP sorted cells is good evidence). However, the stain is hazy and does not look to be specific to nuclei which would be expected for this transcription factor. The authors should provide better images and co-label with DAPI to ensure that the staining is real. This is important as this underlies the point that Oct4+ cells are not actively cycling.

5. Figure 1g. The authors state that Oct4 is a post-proliferative phenomenon. The first piece of evidence for this is that Oct4 ab does not overlap with BrdU (see point 4) The second piece of evidence is that Oct4+ cells via ISH, overlap with PCNA but not BrdU. Since PCNA labeling extends after cell cycle exit and co-labels with Oct4 ISH staining the authors conclude that Oct4 is induced in post-proliferative cells. This is a difficult phenomenon to investigate and I don't think the evidence shown warrants the conclusion made. Lack of Oct4 mRNA signal via ISH does not mean these cells do not have the Oct4 protein and I do not think the antibody labeling is robust enough to trust that these cells lack Oct4 (see point 7)

6. Figure 3 B,D,F. - The suggestion is that Oct4 regulates MGPC formation by controlling cdh1, snails, and Zeb1. Are these factors located in Muller glia? These factors should be located to MGPCs either via ISH, ab staining, or PCR on tuba1a-GFP sorted cells.

7. Figure 4B,E. The antibody labeling to Oct4 is concerning. As mentioned above this is a transcription factor that should be restricted to nuclei. The pattern looks to be everywhere and possibly excluded from the cell nuclei. Labeling is shown all throughout the inner plexiform layer (where dendrites reside and nuclei are excluded). Furthermore, in the control condition (4dpi GFP-mRNA injected) there is no Oct4 labeling in the retina. This is in contrast to Figure 1F where the authors show Oct4+ cells throughout the 4dpi retina. I would be hesitant to trust any conclusions made with this antibody staining.

8. Figure 5 H and Fig 6K Western blot is not convincing by the eye and is not quantified.

9. Figure 6M & N - Looking at BrdU+ cells and which retinal layers they reside is not informative to cell fate. After injury, Muller glia are not the only cells that divide, microglia and endothelial cells will also be BrdU+ and can reside in the INL and GCL. These BrdU+ cells could be neurons, but they could also be persistent progenitors, Muller glia, microglia, endothelial cells or astrocytes. Therefore, the figure does not indicate "functional" regeneration. The extended figure EV6B tries to address these concerns, but no quantifications are done using cell type specific markers. The GS and PKC labeling is very low quality and one is unable to tell if co-localization is occurring. Quantifying the extent of BrdU/HuD+ cells would be helpful to convince readers that these expanded MGPCS following OCT4 knockdown are functional.

10. Discussion: The authors implicate TGFB-signaling as a potential mechanism controlling MGPC formation downstream of Oct4. TGFB-signaling has been previously implicated in MGPC formation in the fish, chick, and mouse. These works should be cited pertaining to that discussion (Close et.al 2005; PMID:15944186, Lenkowski, et al. 2013 PMID: 23918319, Todd, et al. 2017 PMID 28703293)

Referee #3 Review

Report for Author:

The paper by Sharma and colleague seeks to investigate how Oct4 regulates the reprogramming of Muller glia into proliferating progenitors in zebrafish retinas. Evidence is provided that knock-down of Oct4 suppresses reprogramming of Muller glia early after injury, and supports cell cycle exit later in the regenerative response. Evidence is provided that Oct4 interacts with Ascl1a to initiate the process of reprogramming. Evidence is provided that Oct4 may act to promote TGF/Smad later in the process of regeneration to suppress proliferation and promote differentiation. Evidence is provided that over-expression of Oct4 at lower doses, but not higher doses, stimulates proliferation in injured retinas. There is a large volume of effort and data in this manuscript. In too many instances the quality of the data and interpretation fall short.

Many observations are over-extended. For example, although claims of regulation are made, there is no direct evidence for Oct4-mediated regulation in Muller glia during reprogramming. Correlations are suggested by the data, but cell-level mechanisms are not unambiguously demonstrated.

Electroporation does not specifically target Muller glia and appears to cause widespread retinal damage. RT-PCR, qRT-PCR and Western blots from retinal extracts do not show direct regulation of gene expression by Oct4 in discreet populations of cells. However, the data are interpreted as cell-autonomous effects.

In general the quality of the tissues and fluorescence imaging is poor.

The localization of Oct4 and BrdU/PCNA is not compelling. It is not evident why patterns of Oct4 expression with markers of reprogramming were not determined at 16hpi when levels of mRNA and protein appear to peak.

Figure 1k - the co-labeling for Oct4 and tuba1a-GFP is poor.

A hypothesis is posed that Oct4 initially (0-16hpi) acts to suppress reprogramming. Does electroporation (which damages the retina) of Oct4 stimulate the formation of proliferating progenitors?

Figure 4 - the GFP-fluorescence from control and Oct4 over expression appears to be ubiquitous or widespread background. Further, the quality of the tissues and labeling is poor. The BrdU/PCNA-positive cells in 4E with the highest dose of RNA appear to be abnormal. Are these cells dying?

Figure 5 - emerging scRNA-seq zebrafish databases indicate that TGFb-related genes, receptors, ligands and signal-transducers are predominantly expressed retinal neurons, not Muller glia, with the exception of TGFB3, which is expressed at high levels in Muller glia and is absent in proliferating progenitors.

Figure 5 - Why are MOs for Oct4 applied twice? What is the expected half-life for the MOs? Is there any reason to believe that a second delivery of Oct4-MO is required? What are the temporal dynamics of

Figure 6 - the profiles of the nuclei of the BrdU-labeled cells look abnormal. Are these cells pyknotic or perhaps proliferating microglia?

Figure 6E - the tissue quality appears extremely poor. The images may not be representative...?

Numbers of BrdU/Edu+ cells are approximately equal, not 3-fold or 7-fold increased as indicated by the histogram in panel F. Further, the appears to be cellular debris or fragments that are co-labeled for BrdU and EdU. Similarly, in 6m the BrdU labeled cells look like debris, pyknotic nuclei or debris.

Minor point:

- It would be helpful to indicate at the beginning of the Results that injury was induced by a focal stab injury.

Referee #1:

**Comment 1:**

This excellent piece of work shows that Octamer-binding transcription factor 4 (Oct4) is an essential for
reprogramming of Müller cells into a proliferating population of progenitors.

**Answer:**

We thank referee #1 for finding our study that shows the importance of Oct4 in Muller glia reprogramming
as an excellent piece of work.

**Comment 2:**

The paper highlights that several regeneration-associated factors (including Ascl1a, Lin28a, Sox2, Zeb,
E-cadherin, and various microRNAs) are targets for Oct4 regulation.

**Answer:**

You are right. The Oct4 is playing regulatory role on a multitude of regeneration-associated factors

**Comment 3:**

The experiments were carried out with state-of-the art technologies and the manuscript is well written.

**Answer:**

We thank referee #1 for finding the techniques and experimental approach in this study appreciable. We
also thank referee #1 for finding the manuscript well written.

**Comment 4:**

I have no major points of criticism, which is a very rare occasion in my experience as a referee.

**Answer:**

We thank referee #1 for finding the story clean and hence probably you had no major criticism. This
opinion of referee #1 is indeed very exciting to our research group.

**Comment 5:**

As minor point, the term Müller cell should be written as "Müller" and not "Muller". The cell is named
after the german scientist Heinrich Müller, who discovered the cells in the 1840s.

**Answer:**

We welcome this comment. We agree with referee #1 that Muller glia should be written as Müller glia.
We indeed made changes wherever applicable in the revised manuscript. We thank referee #1 for
pointing this out.

-----
Referee #2:

**Comment 1:**

This work by Sharma, et al. implicates the pluripotency gene Oct4 in retinal regeneration in the zebrafish
model. In this model, Muller glia cells respond to a stab injury by reprogramming into Muller glia-derived
progenitor cells (MGPCs) that can regenerate retinal neurons. The authors posit that Oct4 plays a
critical and diverse role in different phases of regeneration. In early phases after injury, the authors
convincingly demonstrate that Oct4 is necessary for MGPC induction and that this probably occurs
through modulation of various regeneration associated genes as well as an ascl1a regulatory loop. At
later time points the authors suggest Oct4 switches to a transcriptional repressor by associating with
Hdac1 and NurD complex and this switch now controls cell cycle exit and repression of regeneration
associated genes.

**Answer:**

We are glad to find that referee #2 finds the importance of Oct4, in causing the induction of MGPCs,
convincing from our study. Referee #2 has rightly pointed out that the Oct4 indeed play a dual role
during early and late phases of regeneration.

**Comment 2:**

The function of Oct4 in retinal regeneration has yet to be explored in the field of retinal regeneration and
the authors suggest a novel and interesting biphasic mechanism. However, many conclusions drawn do
not follow from the data shown and further experiments are needed to provide insight into the
mechanisms downstream of Oct4.

**Answer:**

Indeed, referee #2 in right that the importance of Oct4 was never demonstrated in retinal regeneration
context. We welcome all the critical comments of referee #2, and have carried out additional
experiments, as suggested by referee #2, which would shed light onto mechanisms of gene regulation
downstream to Oct4.

**Comment 3:**

Major point 1. (Figs 3 and 4) The proposed mechanism for how Oct4 controls MGPC induction involves
downregulation of TGFB components, activation of Zeb1/2 and E-cadherin and regulation of miR-200a/b
and mir143/145. However, this is not sufficiently shown. At numerous points throughout the manuscript
language is used such as "it is logical to believe that Oct4 expression indeed upregulates Zeb genes
expression to prevent further reprogramming (line 290)". The authors do provide evidence that these
above listed factors are induced after injury and contain Oct4 binding sites occupied by Oct4. Does
manipulation of Zeb1/2, E-cad, Snails, or the miR's mentioned actually impact regeneration? As it
stands these are correlative phenomenon (mRNA is induced and binding sites are occupied) and do not
imply that these factors control MGPC formation or proliferation.

**Answer:**

We welcome this comment. Yes, the role of Zebs, E-cadherins, miR-200a/b and mir143/145 was not
included because there may not be adequate space in the manuscript figures. We also feared that these
could take away the main focus of the paper that is importance of Oct4-mediated gene regulatory
network. However, in the revised manuscript we have included data to answer these questions from the
referee #2. We apologise for making the figures a bit more dense in the revised manuscript owing to
additional data to support referee's comments.

Referee #2 says that----- "At numerous points throughout the manuscript language is used such as
"it is logical to believe that Oct4 expression indeed upregulates Zeb genes expression to prevent further
reprogramming (line 290)"-----Kindly note that we did not use such a sentence in numerous
places. Except the place where referee #2 has quoted, we found just one more place in the following
quoted sentence from the previous manuscript draft "As we have already found increased levels of *snail*
gene family members namely *snai1a*, *snai1b*, *snai2* and *snai3* and Tgf- β signaling components such as
*tgfr1b*, *tgfb2*, and its effector genes *tgfb1* and *smad7* in *oct4* knockdown background (Figures 3B and
3C), it was logical to examine the levels of these genes in *oct4*-overexpressed retina at 2dpi (line 367 of
old version manuscript)". We do not understand why this sentence was wrong in this given context.
However, we have removed the word 'logical' from both these sentences in the revised manuscript. We
did not find a 3rd place in the entire old manuscript with the usage of 'it was logical to.....'. Again, the
concern raised by referee #2 regarding the quoted sentence also has been validated experimentally in
the revised manuscript, along with the data inclusion in Figure 4.

In other words, we have done the manipulation of Zeb1/2, E-cad, Snails, and the various microRNAs,
dealt with in this study, independently during regeneration, and the data is included along with additional
description in the text of revised manuscript. A few of such manipulations are done along with *oct4*
knockdown as per the requirement in a given context. The entire sets of data is included as a new figure
that is Figure 4. It is also to be noted that the Figure 4 of old manuscript is Figure 5, and so on and so
forth until Figure 8, which was figure 7 in the old manuscript.

**Comment 4:**

Major point 2: The authors propose an interesting biphasic effect where Oct4 promotes MGPC formation
early after injury while at later time points it represses reprogramming genes to induce cell cycle exit.
More data would be needed to make this a compelling case. The first set of data provided that Oct4 has
opposing effects is not based on "regenerative time" but rather on overexpressing Oct4. The authors
show when Oct4 is overexpressed (only at high levels) it switches to a repressive function. They link this
to "regenerative time" by doing a co-IP with the repressive factor Hdac1 and find that these two factors
form a complex at later time points in regeneration but not earlier. This story between Oct4 levels and
phase of regeneration is confusing. Oct4 expression naturally peaks at earlier time points after injury,
yet that's when it has the least affinity to Hdac1. So in the normal regenerative context the affinity of
Oct4 to Hdac1 does not seem to be related to the levels of Oct4. So why do the levels of Oct4 matter
when overexpressed at later time points? Does Oct4 switching from an activator to a repressor depend
on the levels of Hdac1 or NurD? And if so, if you overexpressed HDAC1 early after injury would you
then prematurely switch Oct4 from an activator of regeneration genes to a repressor?

**Answer:**

We welcome the comments and concerns raised by referee #2. It could partly be because of lack of
adequate description in the first version of the manuscript. We have made manuscript more descriptive
in relevant places to make the scenario more clear.

Please note that Oct4 has a unique expression pattern during different stages of retina regeneration.
While it is expressed panretinally during early phases, it stays secluded from actively proliferating

MGPCs (BrdU+ cells seen from the BrdU pulse labelling experiments). When we overexpress Oct4,
pattern of expression of Oct4 is also affected. When transfected with lower concentrations of *oct4*
mRNA, the proliferation is favoured. This could be because, in these scenario, the dilution of *oct4*
mRNA can occur towards the later stages of regeneration and could still retain the 'normal absence' of
Oct4 from the actively proliferating cells, as seen in any regenerating wild-type retina. However, when
we overexpress Oct4 at extremely high concentration, the dilution would NOT happen effectively. In
other words, at high Oct4 concentrations, even the MGPCs would contain *oct4* mRNA (and hence
protein), which is unwanted/undesirable in them. This could potentially decrease the proliferative cell
number as seen in transfection with higher concentrations of *oct4* mRNA. So rather than thinking that
high concentration of Oct4 causes gene repression, it would be more appropriate to think an 'out of
place' expression of Oct4 (especially in high dose of *oct4* mRNA into retina) could be the cause of
reduced MGPCs.

Regarding the affinity of Oct4 for Hdac1, the time course Co-IP between Oct4-Hdac1 as given in Figure
5F (old version it is Figure 4F), reveals that there is an increased Oct4-Hdac1 collaboration towards the
end of retina regeneration. This suggests that there is a great need of gene repression towards late
phases than in the early stages of retina regeneration. However, as one can see from the same Figure
5F that there is ample Oct4 affinity for Hdac1 at 15hpi as well which is effective in causing gene
repression of miR-200a/200b and miR-143/145, as shown by ChIP assay in Figure 3J and 3K of old as
well as revised manuscript. From these data, we could assume that at a given regenerative time, the
functionality of Oct4 is not necessarily to cause exclusive gene activation or repression. It has to be
context dependent. For example in Figure 3G, Oct4 binds to *zeb1a/1b/2a/2b* promoters for their
activation at 16hpi. Similarly, in Figure 7C and 7D, the Oct4 binds to promoters of *hdac1* (Figure 7C)
and *chd4a* (Figure 7D) for an activation cause both at 16hpi and 5dpi. Hence, it would be the right way
to presume that the binding of Oct4 on its target sequence is context dependent, which warrants the
possibility of other decisive partner proteins along with Oct4 in choosing a target and subsequent
activation/repression effects.

To specifically address the questions concerning levels of *hdac1* and its impact on Oct4, raised by
referee #2, we did additional experiments involving overexpression of *hdac1* mRNA. Overexpression of
*hdac1* mRNA abolished its affinity for Oct4 at 15hpi, as revealed in the co-immunoprecipitation assay.
The data is described in Figure 5G. More to this, we also saw an accelerated cellular proliferation in
Hdac1 overexpressed retina at 2dpi. This data is given in Figures EV4C and EV4D. We could presume
that the Oct4-Hdac1 association may be crucial in restricting the number of proliferating cells within
certain limit. As we have already shown that *miR-200a/b* and *miR-143/145* genes get repressed by
Oct4-Hdac1 complex, and this association is lost because of Hdac1 over expression, which could lead
to enhanced cellular proliferation at 2dpi. Referee #2 is right in presuming that Hdac1 overexpression
may have an effect, however the affinity of Oct4 for Hdac1 is definitely not dose-dependent. You may
also note that in Figure 5F, the Oct4 affinity for Hdac1 is high at 4 and 7dpi despite having lower
concentrations of Hdac1, compared to earlier time points.

**Comment 5:**

Other concerns.

3. Figure 1c. The authors state that Oct4 is first induced panretinally and then restricted to the injury site
at later stages post injury. The images for this conclusion are unclear at the later stages (4dpi and 7dpi)
the ISH signal is present across the whole field of view presented. Images with a wider field of view
would be more convincing.

**Answer:**

We welcome this comment, and referee #2 is right. We have selected images that eliminate this
concern, and included in the revised manuscript's Figure 1C.

**Comment 6:**

4. Figure 1f. The pattern of staining for the Oct4 antibody is unconvincing. I do believe that Oct4 is in
MGPCs (the signal in tuba1aGFP sorted cells is good evidence). However, the stain is hazy and does
not look to be specific to nuclei which would be expected for this transcription factor. The authors should
provide better images and co-label with DAPI to ensure that the staining is real. This is important as this
underlies the point that Oct4+ cells are not actively cycling.

**Answer:**

We welcome this comment. However, we wish to clarify a few things here. Please note that BrdU
pulsing for a very short time, marks the cells that are actively cycling during a period of 3hours before
harvesting the retina. However, PCNA and GFP in *1016tuba1a:GFP* transgenic fish stay for longer
duration. Because of this, in actively proliferating (BrdU+) cells, the *oct4* mRNA *in situ* hybridization
signal is weak or nil (Figure 1D). Similarly, the Figure 1G panel shows that *oct4* mRNA *in situ*
hybridization signal is weak or nil in BrdU+ cells but *oct4* expression is co-expressed with PCNA, in the

same cross section with triple staining. Furthermore, the GFP⁺ cells are indeed PCNA⁺ (Figure EV1C),
and these GFP⁺ cells have Oct4 expression too (Figure 1F). The concern of referee #2 regarding the
Oct4 immunostaining is addressed and we have provided a more convincing picture in revised
manuscript in Figure 1F along with DAPI counterstain (also see Figure 5D topmost section).

**Comment 7:**

5. Figure 1g. The authors state that Oct4 is a post-proliferative phenomenon. The first piece of evidence
for this is that Oct4 ab does not overlap with BrdU (see point 4) The second piece of evidence is that
Oct4⁺ cells via ISH, overlap with PCNA but not BrdU. Since PCNA labeling extends after cell cycle exit
and co-labels with Oct4 ISH staining the authors conclude that Oct4 is induced in post-proliferative cells.
This is a difficult phenomenon to investigate and I don't think the evidence shown warrants the
conclusion made. Lack of Oct4 mRNA signal via ISH does not mean these cells do not have the Oct4
protein and I do not think the antibody labelling is robust enough to trust that these cells lack Oct4 (see
point 7)

**Answer:**

We welcome the comment. It is indeed hard to discriminate actively proliferating cells from post-
proliferative cells. However, please note that we made the statement based on quantitative data from
cell counts of *oct4* mRNA expression in BrdU⁺ or PCNA⁺ cells. You may kindly note the time course
expression of *oct4* given in Figures EV1A and EV1B suggest that there is increased co-localization of
*oct4* with BrdU towards late phases of regeneration, when more cells are exiting cell cycle. In other
words at 4dpi, the number of MGPCs are high and they continue to be in cell cycle (which have little
*oct4* mRNA), while at 8dpi the number of MGPCs are much less (which have *oct4* mRNA), and are
quitting the cell cycle. During the late stages, despite having very few BrdU⁺ cells, the propensity of
them having *oct4* expression is high. To make this fact clearer, we have discussed this in detail, in the
revised manuscript. We agree with referee #2 that lack of *oct4* mRNA do not confirm absence of Oct4
protein from a mother cell. However, we request referee #2 to note that, we haven't taken any extra
claim on the functional dynamics of Oct4, throughout the manuscript, solely based on its differential
expression in actively proliferating/post-proliferated cells. All qPCR or ChIP experiments are done in
total retina. Transfection and gene knockdowns are done without any cellular bias on whole retina.

**Comment 8:**

6. Figure 3 B,D,F. - The suggestion is that Oct4 regulates MGPC formation by controlling *cdh1*, *snails*,
and *Zeb1*. Are these factors located in Muller glia? These factors should be located to MGPCs either via
ISH, ab staining, or PCR on *tuba1a*-GFP sorted cells.

**Answer:**

We welcome the comment. It was indeed a valid doubt of ours too. In the revised manuscript, we have
now included these results obtained by qPCR from sorted cells from *1016tuba1a*:GFP transgenic fish.
The data is included in the Figure EV 3G, EV3H, EV3J and EV3N.

**Comment 9:**

7. Figure 4B,E. The antibody labeling to Oct4 is concerning. As mentioned above this is a transcription
factor that should be restricted to nuclei. The pattern looks to be everywhere and possibly excluded from
the cell nuclei. Labeling is shown all throughout the inner plexiform layer (where dendrites reside and
nuclei are excluded). Furthermore, in the control condition (4dpi GFP-mRNA injected) there is no Oct4
labeling in the retina. This is in contrast to Figure 1F where the authors show Oct4⁺ cells throughout the
4dpi retina. I would be hesitant to trust any conclusions made with this antibody staining.

**Answer:**

Please note that Figure 4 is Figure 5 in revised version.

Referee#2 has mentioned concern about the Oct4 antibody immunofluorescence of panels (Figures
1F,4B and 4E), (referee #2's original comments numbers were #4, #5, and #7). Similar to mentioned in
all previous places, once again we state that, we replaced the old Oct4 immunofluorescence with a
clearer, convincing immunofluorescence of Oct4. Similarly, the Figure 4B experiments were repeated
and new data is included in revised manuscript's Figure 5D. However, the content from Figure 4B is
appended onto topmost part of Figure 5D in revised manuscript. The Oct4 immunofluorescence in *gfp*-
mRNA transfected, 4dpi retina is shown along with BrdU and PCNA co-labelling and its merge in Figure
5D of revised manuscript. This image now could clarify the nuclear localization of Oct4 as well.

In Figure 4E, because of overexpression of *oct4* mRNA, there could be varying degree of staining for
the Oct4 protein. Although, because of its elaborate cytoplasm, Muller glia cells can be the major targets
of the transfected mRNA, there can still be uneven density of Oct4 protein at different locations. When
the protein levels go high because of transfection, there can also be cytoplasmic Oct4. Although, native
Oct4 expression could stay restricted to the nucleus, transfected expression of Oct4 need not follow the
usual trend because of molecular crowding. We could say this because in all our repeated transfection
experiments, after Oct4 over expression, the protein can be seen away from nucleus as well. We also

wish to quote another similar scenario herewith. Please note that in zebrafish Muller glia, after a retinal injury, there exists both nuclear and cytoplasmic stabilized beta catenin, a protein that usually localizes to nucleus (*Proc Natl Acad Sci U S A* 108: 15858-63).

Comment 10:

8. Figure 5 H and Fig 6K Western blot is not convincing by the eye and is not quantified.

Answer:

Figure 5 is 6 and Figure 6 is 7 after revision.

We welcome the comment. We included the densitometry plots for Figure 5H and 6K in revised manuscript. The quantifications are done by densitometry and the data are provided in Figures 6I and 7L.

Comment 11:

9. Figure 6M & N - Looking at BrdU+ cells and which retinal layers they reside is not informative to cell fate. After injury, Muller glia are not the only cells that divide, microglia and endothelial cells will also be BrdU+ and can reside in the INL and GCL. These BrdU+ cells could be neurons, but they could also be persistent progenitors, Muller glia, microglia, endothelial cells or astrocytes. Therefore, the figure does not indicate "functional" regeneration. The extended figure EV6B tries to address these concerns, but no quantifications are done using cell type specific markers. The GS and PKC labeling is very low quality and one is unable to tell if co-localization is occurring. Quantifying the extent of BrdU/HuD+ cells would be helpful to convince readers that these expanded MGPCS following OCT4 knockdown are functional.

Answer:

Figure 6 is Figure 7 after revision. The figure EV6B is put in Appendix Figure S1C and S1D.

We welcome the comment. In Figure 6M and N, the sole purpose was to check if the increased MGPCs seen because of late *oct4* knockdown stay back after 30 days of regeneration. In other words, we wished to examine, if the increased MGPCs formed in late *oct4* knockdown retina are viable. The results indicate that the cells are indeed viable and all three layers (ONL, INL and GCL) had an increased number of cells compared to control sets. We have toned down the word "functional" to make the point clear that these cells are capable of migration and differentiation to retinal cell types. We now have included quantitative data on various cell-types, as well as more convincing/representative immunofluorescence image panels in the revised manuscript and included as Appendix Figure S1C and S1D. We agree to referee#2 comments that endothelial cells, microglia etc also can take up BrdU, but the injury is uniform both in control MO and *oct4* MO treated conditions. So the comparison is always made to respective control experiments. In other words, any cell-type (Muller glia, micro glia, astrocyte, endothelial cell) that take up BrdU (7-9dpi) and stay back upto 30 days post injury (a time when cellular differentiation is almost complete) could be detected. However, any non-Muller glia cellular proliferation must have to be similar between control MO and *oct4* MO electroporation experiments. In this scenario, if we found an enhanced number of BrdU+ cells in 30dpi retina, because of *oct4* knockdown (compared to control), it indicates that the enhanced number of MGPCs seen in *oct4* late knockdown experiments indeed are viable and localized to various layers. The goals of the Figures 7M-O are solely to convey this.

Comment 12:

10. Discussion: The authors implicate TGFB-signaling as a potential mechanism controlling MGPC formation downstream of Oct4. TGFB-signaling has been previously implicated in MGPC formation in the fish, chick, and mouse. These works should be cited pertaining to that discussion (Close et. al. 2005; PMID:15944186, Lenkowski, et al. 2013 PMID: 23918319, Todd, et al. 2017 PMID 28703293)

Answer:

We welcome this comment. We thank referee #2 for the suggestion to include these references. We have made the changes accordingly in the revised manuscript's discussion.

Referee #3:

Comment 1:

The paper by Sharma and colleague seeks to investigate how Oct4 regulates the reprogramming of Muller glia into proliferating progenitors in zebrafish retinas. Evidence is provided that knock-down of Oct4 suppresses reprogramming of Muller glia early after injury, and supports cell cycle exit later in the regenerative response. Evidence is provided that Oct4 interacts with Ascl1a to initiate the process of reprogramming. Evidence is provided that Oct4 may act to promote TGF/Smad later in the process of regeneration to suppress proliferation and promote differentiation. Evidence is provided that over-expression of Oct4 at lower doses, but not higher doses, stimulates proliferation in injured retinas. There is a large volume of effort and data in this manuscript. In too many instances the quality of the data and

335 interpretation fall short.

**Answer:**

We thank referee #3 for the analysis of the major findings, and finding the study robust. We were happy
to address all your queries and addressed them in detail in revised manuscript with new data.

**Comment 2:**

Many observations are over-extended. For example, although claims of regulation are made, there is no
direct evidence for Oct4-mediated regulation in Muller glia during reprogramming. Correlations are
suggested by the data, but cell-level mechanisms are not unambiguously demonstrated. Electroporation
does not specifically target Muller glia and appears to cause widespread retinal damage. RT-PCR, qRT-
PCR and Western blots from retinal extracts do not show direct regulation of gene expression by Oct4 in
discreet populations of cells. However, the data are interpreted as cell-autonomous effects.

**Answer:**

We would be happy to clarify and modify over-extended claims, if any. Kindly note that Muller glia
reprogramming is a well-accepted fact across different models (fish, mice, chicken etc) of retina
regeneration, and we have cited a large number of such studies in the manuscript. Further, the evidence
of Muller glia reprogramming as seen in models such as zebrafish is mainly obtained through the
description of the outcome of Muller glia reprogramming such as cell proliferation, prior to regeneration.
This is often done mainly because of unavailability of a *bona fide* cellular marker that initiate and restrict
its expression exclusively in reprogrammed Muller glia. In this scenario, no one could claim that a given
Muller glia is reprogrammed and the other one is not. In other words, the extent/efficiency of Muller glia
reprogramming can be measured ONLY through the quantification of the progenitor cells (MGPCs) in the
injured retina at various time points post retinal injury. I would also like to mention here that in the 30
gauge needle-stab injury model, the proliferation begins at 36-48 hour post injury and peaks at 4 days
and almost completely stops at 7-8 days post injury (*J. Neuroscience*, 26: 6303-13). Based on this we
have chosen a time for various gene expression analysis.

Cell-level data is a wonderful option in these scenario provided there are adequate tools available in
zebrafish model. Unfortunately, such tools are limited. Kindly note that we haven't taken any claim that a
given result is cell-type specific, unless a strong evidence is given to state it. We have demarcated cells
based on BrdU/PCNA/GFP expressions, mainly. There are ample literature (which are cited too) which
state that in zebrafish retina, it is the Muller glia that contributes to progenitor formation in various injury
models. It has been proven through genetic labelling and lineage tracing studies as well (*J.*
*Neuroscience*, 26: 6303-13; *J Comp Neurol* 518: 4196-212). Furthermore, the focus of this study is to
understand the importance of Oct4 induction in injured retina and subsequent gene expression events
rather than restricting to a given cell-type of the retina. It is also important to note that a given cell's
response also depends on neighbouring cell's physiology as well, especially in an *in vivo* system.

Morpholino electroporation into retina is done in various research studies published in reputed journals
from various research groups. I wish to clarify a few things here that (a) Electroporation (as per the
described, published parameters using a given electroporator) does not damage retina, as evident from
absence of spontaneous cell proliferation/apoptosis in uninjured retina. (b) Electroporation of standard
control morpholino does NOT affect the retina regeneration. (c) There are several gene knockdown
examples that can accelerate or decelerate the MGPCs proliferation, purely based on the gene in
question that is being knocked down. Based on these three facts one shouldn't presume that
electroporation causes wide spread retinal damage. We do not claim that morpholino electroporation
targets the Muller glia specifically. It can enter any cell, however the knockdown can happen only if the
gene which is being targeted has its expression in that cell. In other words, if a given MO entered a cell
which do not have the corresponding mRNA target, it doesn't do any harm to that cell or block some
other random mRNA targets. Please note that Muller glia cytoplasm is more spread out and wraps
around other neurons, moreover, approximately 40% of retinal cells are Muller glia. These factors give
the Morpholino better access into Muller glia cytoplasm than any other cell types of the retina.

Referee #3 is right in pointing out that none of the phenomenon based on gene knockdown are meant
only for a given cell type. However, if a gene is expressed only in Muller glia and if a morpholino is
delivered against that gene in every cell, the knockdown happens only in Muller glia, as other cells do
not express this gene to be knocked down. All our RT-PCR, qRT-PCR and Western blots from retinal
extracts represent holistic phenomena observed in retina. Kindly note that we did not claim anywhere
that we selectively knockdown a given gene exclusively in this cell type. If we used *oct4* morpholino in
injured retina, our goal was to find the response of the retina in the complete or near complete absence
of Oct4 protein in the entire retina.

Electroporation does NOT damage the retina. To make this point clear and more convincing, we have
done an experiment in which fish were electroporated as we do for morpholino delivery, then harvested
on day 4 (to match the 4day post injury in routine experiments) to perform both TUNEL (to show if there

is cell death because of electroporation) and PCNA (to show if there is a proliferation because of
 electroporation). This data is included in the rebuttal letter herewith below. Kindly note that, this
 experiment is performed, just to make the point clear that, no damage/regenerative proliferation/cell
 death occur because of electroporation.

Comment 3:

In general the quality of the tissues and fluorescence imaging is poor. The localization of Oct4 and
 BrdU/PCNA is not compelling. It is not evident why patterns of Oct4 expression with markers of
 reprogramming were not determined at 16hpi when levels of mRNA and protein appear to peak.

Answer:

We welcome this comment. We have now modified the immunofluorescence images using a new
 antibody. The data of Oct4 immunofluorescence is given in Figures 1F and 5D. In the revised
 manuscript we have included, alongside the data of *oct4* knockdown at 2dpi, the data for *oct4*
 knockdown at 16hpi for *Ascl1a* and *Sox2* (both mRNA and protein levels) that is discussed as well. This
 data is now appended on to the Figures 2E and 2F.

Comment 4:

Figure 1k - the co-labeling for Oct4 and tuba1a-GFP is poor.

Answer:

We welcome this comment. We have now modified the Oct4 and *1016tuba1a*:GFP immunofluorescence
 images with better ones in the revised manuscript. This data is now included in Figure 1F, and fully
 removed the Figure 1K panel.

Comment 5:

A hypothesis is posed that Oct4 initially (0-16hpi) acts to suppress reprogramming. Does electroporation
 (which damages the retina) of Oct4 stimulate the formation of proliferating progenitors?

Answer:

Kindly note that we do not say anywhere in the manuscript that “Oct4 initially (0-16hpi) suppress
 reprogramming”. In fact Oct4 is necessary to push the MG cells to reprogram. There are other studies
 (cited in this manuscript too) in embryonic stem cells (ESC) of mammalian system to demonstrate that
 Oct4 initially push the reprogramming to induce ESCs and then Oct4 can be downregulated several fold
 without losing ESC self-renewal characteristics, and later on the induction of Oct4 pushes the cells into
 differentiation pathways. We feel that referee #3 understood that Oct4 is suppressing reprogramming
 because of the following sentence in the discussion of previous draft..... “This observation was
 intriguing and appears to be the result of an anti-cell reprogramming role of Oct4 during retina
 regeneration”..... It was in context to Oct4 favouring Zeb expression through upregulation of *zeb*
 mRNA and downregulation of its repressor *miR-200a/b* genes. However, we have removed this
 sentence from ‘discussion’ of revised manuscript to avoid such a confusion.

Electroporation conditions we follow (as mentioned in the materials and methods) do NOT damage the
 retina. Please see above where we showed uninjured retina with or without electroporation, along with
 PCNA/DAPI/TUNEL staining. It is to be noted that injury is caused by the needle poke into the retina
 throughout our study. Kindly note that, no damage/regenerative proliferation/cell death occur because of
 electroporation.

Referee #3 says that “ Does electroporation (which damages the retina) of Oct4 stimulate the formation
 of proliferating progenitors?” We have done *oct4* targeting morpholino electroporation, both at early and

453 late stages. Again, we did transfection of *oct4* mRNA in other experiments. Hence the question saying “
 -----electroporation of Oct4-----” is unclear to us. We presume that referee #3 meant to ask if the
 late electroporation of *oct4* morpholino stimulate the formation of proliferating progenitors (MGPCs). The
 answer is, compared to standard control morpholino electroporated retina, there is an increased number
 of progenitors (MGPCs) in *oct4* morpholino late-electroporation (5 to 8dpi window). However, as already
 discussed in detail in the previous version of the manuscript itself, we would like to emphasize here too
 that the increase in progenitors (MGPCs) number is NOT because of new progenitor formation, rather
 the already existing progenitors fail to exit cell cycle. This is clarified with BrdU and EdU labeling
 experiments and cell counts. In other words, the progenitors (MGPCs), that were present at 5dpi (until
 the control morpholino/*oct4* morpholino electroporation was done) continue to be in proliferating phase
 in *oct4* morpholino electroporated condition, but not with control morpholino. Please note that such a
 phenomenon was NOT found with control morpholino suggesting that these observations are because
 of *oct4* knockdown, rather than presuming an effect of electroporation, which was done both in control
 morpholino and *oct4* morpholino electroporated retina.

**Comment 6:**

Figure 4 - the GFP-fluorescence from control and Oct4 over expression appears to be ubiquitous or
 widespread background. Further, the quality of the tissues and labeling is poor. The BrdU/PCNA-
 positive cells in 4E with the highest dose of RNA appear to be abnormal. Are these cells dying?

**Answer:**

Figure 4 is Figure 5 in revised manuscript.

Referee #3 is right. Please note that the overexpression experiments are done using *in vivo*
 transfection. There is no bias for the entry of liposomes into a given cell. This makes the appearance of
 GFP ubiquitous and widespread. We have replaced the highest dose (Figure 4E) with better one and
 the data is given in Figure 5D. Also, it is important to note that whenever there is a reduced proliferation
 because of lack of proper regenerative response, the BrdU/PCNA cells do not seem robust and do not
 form the typical clustering pattern. This is well documented in various studies that are cited in the
 manuscript. So we request you not to expect the same morphology/pattern of BrdU staining in a
 proliferation/regeneration-compromised retina. Please see below the TUNEL assay data. The green
 spots are the potential TUNEL positive cells that are undergoing apoptosis, especially in the INL and
 INL-ONL boundary. There exists some autofluorescence in the ONL.

 **Comment 7:**

Figure 5 - emerging scRNA-seq zebrafish databases indicate that TGFb-related genes, receptors,
 ligands and signal-transducers are predominantly expressed retinal neurons, not Muller glia, with the
 exception of TGFB3, which is expressed at high levels in Muller glia and is absent in proliferating
 progenitors.

**Answer:**

Figure 5 is Figure 6, after revision.

We welcome the statement of referee #3. We do not associate the expression of *tgfb* ligand or its
 receptor to a particular cell-type. Please note that *Tgfb* is a ligand and it stays sequestered in the extra
 cellular matrix of various tissues in its inactive form. This needs to be modified proteolytically to activate

by various means such as action of metalloproteinases. This activated Tgfb (ligand) binds on to their
respective receptors (tgfb2) expressed in the vicinity cells and often induce more of the
metalloproteinases causing a positive feedback loop. We would also like to emphasize here that, we did
not mention in the manuscript that tgfb or its receptors are from a given cell-type. The goal of the tgfb
ligand or tgfb receptor genes' (performed in whole retina) expression analysis was to find out the holistic
response of the retina in a given experimental condition, such as *oct4* mRNA overexpression or its
knockdown. Please note that both these experimental conditions are unbiased (means not targeting to a
given cell-type), and hence the impact of *oct4* over expression/knockdown on tgfb component genes
also need to be seen accordingly.

**Comment 8:**

Figure 5 - Why are MOs for Oct4 applied twice? What is the expected half-life for the MOs? Is there any
reason to believe that a second delivery of Oct4-MO is required? What are the temporal dynamics?

**Answer:**

Figure 5 is Figure 6, after revision.

We believe that referee #3 may have mistaken with our timeline short-cut description panel. We would
like to emphasize here that in none of the experiments, we delivered morpholinos twice. Please note
that if the word "electroporation" is added in the timeline, that is the point in which the morpholino
actually entered the retina. This point is clarified in the revised manuscript's respective figure legends.
Morpholinos are delivered into the vitreous of the retina at the time of injury. But it enters the cells only
when electroporated. May be referee #3 got it wrong that morpholino was delivered twice in late
electroporation experiments. Please note that this fact was mentioned in earlier draft of the manuscript
itself. We have elaborated this to avoid confusion.

Several publications in past two decades have shown that morpholinos have really long half-life. In
zebrafish embryos, published reports say its existence up to 10 days of development. Similarly, in adult
retina it stays more than 30 days. In the vitreous too, the morpholinos stay active and undisturbed. This
is because soon after injury the blood clot formed prevents the escape of any vitreous material out of
the eye. The morpholino stays in the vitreous until electroporated. Again, owing to high molecular weight
(25 bases nucleotides), the morpholino never gets free entry into any cells by simple diffusion. Also,
morpholinos are neither degraded nor pumped out of the cell. It is really stable molecule even at room
temperature for years. The possible inefficacy that can happen with morpholino delivery is only through
dilution effect because of cell division.

Referee #3 says "What are the temporal dynamics?" Although we are not clear about this question's
real meaning, we believe it is regarding the life of morpholinos. The answer is given in above paragraph.

**Comment 9:**

Figure 6 - the profiles of the nuclei of the BrdU-labeled cells look abnormal. Are these cells pyknotic or
perhaps proliferating microglia? Figure 6E - the tissue quality appears extremely poor. The images may
not be representative...? Numbers of BrdU/EdU+ cells are approximately equal, not 3-fold or 7-fold
increased as indicated by the histogram in panel F. Further, the appears to be cellular debris or
fragments that are co-labeled for BrdU and EdU.

**Answer:**

Figure 6 is Figure 7, after revision.

We have replaced these image panels with better ones that show wider area. These cells are not
pyknotic, as there is no cell death revealed in TUNEL assay. We would like to emphasize here that the
BrdU was delivered at 5dpi and if the cell continue to be in cell-cycle (because of late *hdac1* knockdown
as discussed in Figure 7), there would be a dilution effect on the BrdU because of repeated cell
divisions. This could make the BrdU staining dilute, non-uniform and a bit abnormal in some cells.

Coming to the other point raised, even if they are proliferating microglia, the inference from this
experiment was that the late knockdown of *hdac1* (5 to 8dpi) force the already proliferating cells
(irrespective of the source of origin) to continue in cycling stage. This is evident from BrdU-EdU co-
labelling at 8dpi. Any other parameters such as, a different cell-type that started proliferation or
underwent apoptosis (if any), get normalized between the control MO and *hdac1* MO electroporated
sets.

In this Figure 6E panel (Figure 7E after revision), the experiment done was labeling the MGPCs with
BrdU at 5dpi and then electroporated with *hdac1* morpholino (which was delivered at the time of injury,
but stayed in vitreous and have not entered the retina, until electroporated) to allow the entry of
morpholinos into retina, then given a EdU pulse on 8dpi 3 hours before harvesting. This is well-
described in manuscript as well. In other words, because of *hdac1* knockdown, the BrdU labeled
MGPCs continue to be in proliferative phase evident from the EdU co-labeling. The observation of
referee #3 that number of BrdU+/EdU+ cells are approximately equal is the right observation and we

also wanted to show that only from this result. Again, referee #3 says "BrdU+/EdU+ cells are
approximately equal and not 3-fold or 7-fold increased as indicated by the histogram in panel F"
However, the figure 6F (Figure 7F after revision) panel describe the BrdU+/EdU+ co-labeled cells (kindly
note the label on Y-axis) that increase in MGPCs number because of *hdac1* knockdown and compared
to control morpholinos. We believe that referee #3 thought that we are saying there is 3-fold or 7-fold
increase in EdU+ cells compared to BrdU+ ones. It is absolutely a wrong interpretation which we do not
mention in the manuscript text or figure legend.

However, in case referee #3 is meaning that in all panels the co-labelled BrdU/EdU+ cells look equal in
number, we have replaced these figure panels (Figure 6E) with wider panels in the revised manuscript
to convey the message better. This data is included in Figure 7E. We apologize for the narrow panels in
Figure 6E in the earlier version of the manuscript.

Please see below the TUNEL assay data. The green spots are the potential TUNEL positive cells that
are undergoing apoptosis, especially in the INL and INL-ONL boundary. There exists some
autofluorescence in the ONL.

**Comment 10:**

Similarly, in 6m the BrdU labeled cells look like debris, pyknotic nuclei or debris.

**Answer:**

Yes, we replaced the panels of Figure 6M as well, to make it more appealing. The new data is in Figure
7N. Please see below the TUNEL assay data. The green spots are the potential TUNEL positive cells
that are undergoing apoptosis, especially in the INL and INL-ONL boundary. There exists some
autofluorescence in the ONL.

**Comment 11:**

Minor point:

- It would be helpful to indicate at the beginning of the Results that injury was induced by a focal stab injury.

**Answer:**

Yes, we did this change.

September 12, 2019

RE: Life Science Alliance Manuscript #LSA-2019-00548-T

Dr. Rajesh Ramachandran
Indian Institute of Science Education and Research, Mohali
Department of Biological Sciences
Knowledge City, SAS Nagar
Sector 81
Mohali, Punjab 140306
India

Dear Dr. Ramachandran,

Thank you for transferring your revised manuscript entitled "Oct4 mediates Müller glia reprogramming and cell cycle-exit during retina regeneration in zebrafish" to Life Science Alliance. Your manuscript was reviewed twice before at another journal and the editors transferred those reports to us with your permission.

The reviewers who evaluated your work at another journal appreciated the changes introduced in revision, but overall expected more in-depth insight. Lack thereof is not precluding publication in Life Science Alliance, and we would like to invite you to submit a final version of your manuscript for publication here. Please introduce text changes to address the electroporation / cell death criticism and to facilitate reading. Please also:

- add a summary blurb in our submission system
- indicate in the figure legends which statistical test has been used wherever p values are mentioned

A. FINAL FILES:

B. MANUSCRIPT ORGANIZATION AND FORMATTING:

Sincerely,

Andrea Leibfried, PhD
Executive Editor
Life Science Alliance
Meyerhofstr. 1
69117 Heidelberg, Germany

t +49 6221 8891 502
e a.leibfried@life-science-alliance.org
www.life-science-alliance.org

September 17, 2019

RE: Life Science Alliance Manuscript #LSA-2019-00548-TR

Dr. Rajesh Ramachandran
Indian Institute of Science Education and Research, Mohali
Department of Biological Sciences
Knowledge City, SAS Nagar
Sector 81
Mohali, Punjab 140306
India

Dear Dr. Ramachandran,

Thank you for submitting your revised manuscript entitled "Oct4 mediates Müller glia reprogramming and cell cycle-exit during retina regeneration in zebrafish". We would be happy to publish your paper in Life Science Alliance pending final revisions. Please clearly mention in the methods section that you did not observe cell death in response to electroporation of the retina. Please also note that we do not have EV or appendix figures at LSA. Please rename those to supplementary figure S1 etc. The current Expanded View table should be renamed supplementary table S1, please.

A. FINAL FILES:

-- Summary blurb (enter in submission system): A short text summarizing in a single sentence the study (max. 200 characters including spaces). This text is used in conjunction with the titles of papers, hence should be informative and complementary to the title. It should describe the context

and significance of the findings for a general readership; it should be written in the present tense and refer to the work in the third person. Author names should not be mentioned.

B. MANUSCRIPT ORGANIZATION AND FORMATTING:

Sincerely,

Andrea Leibfried, PhD
Executive Editor
Life Science Alliance
Meyerohofstr. 1
69117 Heidelberg, Germany
t +49 6221 8891 502
e a.leibfried@life-science-alliance.org
www.life-science-alliance.org

September 24, 2019

RE: Life Science Alliance Manuscript #LSA-2019-00548-TRR

Dr. Rajesh Ramachandran
Indian Institute of Science Education and Research, Mohali
Department of Biological Sciences
Knowledge City, SAS Nagar
Sector 81
Mohali, Punjab 140306
India

Dear Dr. Ramachandran,

Thank you for submitting your Research Article entitled "Oct4 mediates Müller glia reprogramming and cell cycle-exit during retina regeneration in zebrafish". It is a pleasure to let you know that your manuscript is now accepted for publication in Life Science Alliance. Congratulations on this interesting work.

*****IMPORTANT:** If you will be unreachable at any time, please provide us with the email address of an alternate author. Failure to respond to routine queries may lead to unavoidable delays in publication.*******

DISTRIBUTION OF MATERIALS:

Again, congratulations on a very nice paper. I hope you found the review process to be constructive and are pleased with how the manuscript was handled editorially. We look forward to future exciting

submissions from your lab.

Sincerely,
